# GPCRome-wide analysis of G-protein-coupling diversity using a computational biology approach

**Marin Matic** [1,3], **Pasquale Miglionico** [1,3], **Manae Tatsumi**[2,3], **Asuka Inoue** [2] ✉ & **Francesco Raimondi** [1] ✉

GPCRs are master regulators of cell signaling by transducing extracellular stimuli into the cell via selective coupling to intracellular G-proteins. Here we present a computational analysis of the structural determinants of G-protein-coupling repertoire of experimental and predicted 3D GPCR-G-protein complexes. Interface contact analysis recapitulates structural hallmarks associated with G-protein-coupling specificity, including TM5, TM6 and ICLs. We employ interface contacts as fingerprints to cluster $G_s$ vs $G_i$ complexes in an unsupervised fashion, suggesting that interface residues contribute to selective coupling. We experimentally confirm on a promiscuous receptor (CCKAR) that mutations of some of these specificity-determining positions bias the coupling selectivity. Interestingly, $G_s$-GPCR complexes have more conserved interfaces, while $G_{i/o}$ proteins adopt a wider number of alternative docking poses, as assessed via structural alignments of representative 3D complexes. Binding energy calculations demonstrate that distinct structural properties of the complexes are associated to higher stability of $G_s$ than $G_{i/o}$ complexes. AlphaFold2 predictions of experimental binary complexes confirm several of these structural features and allow us to augment the structural coverage of poorly characterized complexes such as $G_{12/13}$.

G-protein-coupled receptors (GPCRs) constitute the largest family of cell-surface receptors, making them a primary pharmacological class which is targeted by approximately one-third of the marketed drugs[1]. They transduce extracellular physico-chemical stimuli to intracellular signaling pathways by coupling to one or more heterotrimeric G-proteins[2,3], which are grouped into four major families: $G_s$, $G_{i/o}$, $G_{q/11}$ and $G_{12/13}$ based on homology of their α-subunits[4]. GPCRs' downstream activity is controlled by β-arrestins, which desensitize GPCRs' activity and provide an additional layer of signaling modulation via ERK[5]. Receptor's conformational change upon ligand binding leads to recognition and activation of intracellular G-proteins. Every mammalian GPCR displays a unique repertoire of G-protein-coupling preferences, ranging from highly selective to promiscuous profiles, which

orchestrate specific cellular responses[6]. Aberrant signal transduction is linked to many pathological states, including cancer[7–12]. A mechanistic understanding of the signal transduction processes, integrated with multi-modal data associated with a disease state, can inform targeted therapies and personalized medicine procedures (e.g.[13]).

The experimental profiling of specific coupling preferences is critical to understanding GPCR biology and pharmacology. The binding activities of GPCRs for transducer proteins are being quantitatively screened via medium-throughput methodologies[14–17]. Based on binding profiling from these large-scale experimental assays, sequence-based machine learning for coupling specificity has been proposed[18,19]. Phylogenetic analysis of co-evolutionary patterns inferred from sequence alignments of GPCRs and G-proteins have also provided

[1]Laboratorio di Biologia Bio@SNS, Scuola Normale Superiore, Pisa 56126, Italy. [2]Graduate School of Pharmaceutical Sciences, Tohoku University, Sendai, Miyagi 980-8578, Japan. [3]These authors contributed equally: Marin Matic, Pasquale Miglionico, Manae Tatsumi. ✉e-mail: iaska@tohoku.ac.jp; francesco.raimondi@sns.it

insights into the sequence determinants of coupling specificity, for the entire GPCR family[20] as well as for specific subfamilies[21]. The abundance of experimental structures for GPCRs in ligand-dependent, alternative functional states has also shed light on the structural hallmarks controlling receptor activation for class A GPCR[22] as well as across classes[23].

The determination of receptor-G-protein complex structures is also progressing rapidly, with over 360 complex structures deposited in the PDB (as of March 2023). The determination of the first structures of $G_i$ coupled receptor complexes allowed for initial comparisons with $G_s$ counterparts and highlighted a role of TM5 and TM6 as selectivity filter[24–28]. As a complement to the release of the MT1-$G_i$ complex, we also systematically compared available $G_s$ and $G_{i/o}$ complexes with Class A receptors in terms of interface contact networks and G-protein docking mode similarity assessed via structural alignment[29]. The recent determination of four structures of the serotonin receptors (e.g. 5-HT4, 5-HT6 and 5-HT7 with $G_s$, and 5-HT4 with $G_{i1}$) confirmed the role of TM5 and TM6, and in particular their variable length, as a selectivity filter for G-protein binding[30]. The authors also showed via bioinformatics analysis that this macro-switch is conserved among other class A GPCRs[30]. Yet, a comprehensive picture of the structural hallmarks of coupling specificity remains elusive.

In this work we analyze through structural bioinformatics experimental, as well as predicted, GPCR-G-protein 3D complexes to shed further light on the structural basis of coupling specificity through the analysis of interaction interface contact networks, G-protein docking modes and binding energies (Fig. 1A).

## Results

### Different G-protein complexes are characterized by alternative contact network topologies

We considered a total of 362 3D experimental GPCR-G-protein complexes, comprising 166 $G_s$, 184 $G_{i/o}$, 9 $G_{q/11}$ and 3 $G_{12/13}$ complexes, corresponding to 93, 17, 10, 3, and 2 unique receptors from Class A, B1, B2, C and Frizzled, respectively, and entailing 9 different G-proteins (i.e. GNAS, GNAI1, GNAI2, GNAI3, GNAT1, GNAO1, GNAQ, GNA11, GNA13) (Fig. 1B; Supplementary Data 1). To avoid any bias due to redundant structures solved for the same GPCR-G-protein complex, we derived a set of 125 non-redundant 3D complexes by considering representative structures for each receptor-G-protein pair, using resolution and canonical sequence coverage as selection criteria (Fig. 1C; see Methods). We first identified the residues that are in spatial contact at the GPCR-G-protein interaction interface (see Methods). We then mapped contacting residues to consensus numbering through GPCRdb[31] (Supplementary Data 2) and the common G-protein numbering (CGN)[32] schemes (Supplementary Data 3), respectively for GPCRs and G-proteins. We aggregated contacts based on secondary structure elements (SSEs; Fig. 2A), to yield a network of interacting SSE elements at GPCR-G-protein interfaces (Fig. 1A and Fig. 2A). For the most abundant coupling groups (i.e. $G_s$ and $G_{i/o}$), we derived specific SSE contact networks by pooling contacts on the basis of the bound G-protein. SSE contact networks highlight structural signatures specific to each coupling group. Certain SSEs are invariably central within the interface network, such as TM5, ICL2, or ICL3 for GPCRs (Fig. 2B–D) or H5 for the G-protein (Fig. 2B, C, E). Other elements vary their connectivity based on the bound G-protein. In particular, TM5 has a higher degree of interacting SSEs in $G_s$ complexes as well as an overall number of contacts, while $G_{i/o}$ complexes are instead characterized by higher interconnectivity at the ICL1, TM3, TM6, and ICL3 (Fig. 2B–D). Differences in the overall network topology also emerged when we measured the information flow, quantified as the number of shortest paths passing through each node (i.e. betweenness centrality; see Methods). Indeed, TM5, ICL2, and H8 have a higher betweenness centrality in $G_s$ complexes, while TM3, TM6, and ICL1 prevail in $G_{i/o}$ ones (Fig. 2F). Overall, $G_s$ contact graphs are significantly different from $G_{i/o}$ ones, as

assessed by comparing distances, computed as the Frobenius norm of the difference between the adjacency matrices of the interface contact graphs (permanova $P = 1E-06$; see Methods).

### Contact interface fingerprints imprint coupling specificity

We employed interface contacts to build interaction fingerprints, which are vectors that numerically encode the presence or absence of contact, and which can be used to compare in an unsupervised way GPCR-G-protein complexes based on their interface's structural features (Fig. 1A). We have generated interface fingerprints by mapping either residue pairs at each vector position (Complex fingerprints, or CF), or contact positions separately for the receptor and G-protein (Receptor and G-protein fingerprints, respectively RF and GF; see Methods). We also estimated the contact positions that are more frequent than expected in each coupling group through log-odds ratio statistics (see Methods), and we used this information to filter the most informative contacts for $G_s$ and $G_{i/o}$ couplings (Fig. 3). CF clustering identifies two main clusters: the largest one (cluster 1), is enriched with $G_s$ complexes from both class A and Class B, as well as some $G_{i/o}$ complexes involving class A, class C and F receptors (Fig. 3 and Fig. 4A). The second cluster (i.e. cluster 2), is enriched almost exclusively with $G_{i/o}$ complexes (Fig. 3 and Fig. 4A). In cluster 2, more receptors show promiscuous couplings towards other G-proteins, in particular towards $G_{12/13}$ (35% vs 15% in Cluster 1) and $G_{q/11}$ (55% vs 45% in Cluster 1) (Fig. 4A). The higher promiscuity between $G_{i/o}$ and $G_{12/13}$ couplings is also observed when considering all couplings, even with no structure, available from the Universal Coupling Map (UCM[33]; Supplementary Fig. 1). When relaxing the criteria to perform CF clustering by removing the log-odds ratio filter and by including all unique complexes, the clustering is now mainly guided by the receptor class, with the largest one entailing Class A complexes and the smallest one all the other classes (Supplementary Fig. 2). Within each cluster, subclusters enriched in $G_s$ or $G_{i/o}$ complexes can be identified. Available $G_{12/13}$ and most of the $G_{q/11}$ complexes cluster within the $G_{i/o}$-enriched subcluster (Supplementary Fig. 2). In detail, the S1PR2-GNA13 structure (i.e., PDBID: 7T6B), is clustered within Class A, $G_{i/o}$ subgroup along with other $G_{12/13}$ binders, such as LPAR1 and S1PR5. Similarly, ADGRG1- and ADGRL3-GNA13 complexes are found within the Gi/o subcluster of the second, class B enriched, cluster (Supplementary Fig. 2). This suggests a structural (likely evolutionary imprinted - see Discussion) connection between $G_{i/o}$, $G_{q/11}$ and $G_{12/13}$ proteins.

Overall, $G_s$ couplings are characterized by a significantly higher number of enriched contacts with respect to $G_{i/o}$ ones (Pmann-whitney = 4.69E-14; Fig. 4B). We also performed clustering and enrichment with fingerprints of receptors (RF) and G-proteins (GF) separately. The RF clustering chiefly points to inter-class differences, separating complexes formed by ClassA receptors from those involving other classes (Supplementary Fig. 3A), while not showing particular contact enrichment differences between $G_s$ and $G_{i/o}$ complexes (Pmann-whitney = 4.4E-1; Fig. 4C). The GF clustering better separates these groups (Supplementary Fig. 3B) and displays marginally significant differences in contact enrichment distributions (Pmann-whitney = 4.4E-2; Fig. 4C). This suggests that the combination between G-protein and receptor's residues provides maximum fine-tuning to the recognition process (Fig. 4B).

Complex fingerprints clustering and heatmaps helped visualizing the contact positions that are characteristic of certain G-protein-couplings (Figs. 3A and 4D, E). For instance, several contacts are observed more frequently in $G_s$ complexes, including ICL2-s2s3.1, ICL2.52-G.S3.1, 5.65-G.H5.26, 5.68-G.H5.26, 5.72-G.h4s6.20, ICL3-h4s6.3, 6.39-G.H5.24, 6.39-G.H5.25, 6.40-G.H5.25. The latter positions (i.e. 6.39 and 6.40) are example of $G_s$, class B-specific contacts (Fig. 3 and Fig. 4D, F). These are favored by the characteristic TM6 break characterizing Class B receptors[34], which allows residues on the TM6's N-terminal half to approach the G-protein H5 C-term. On the other

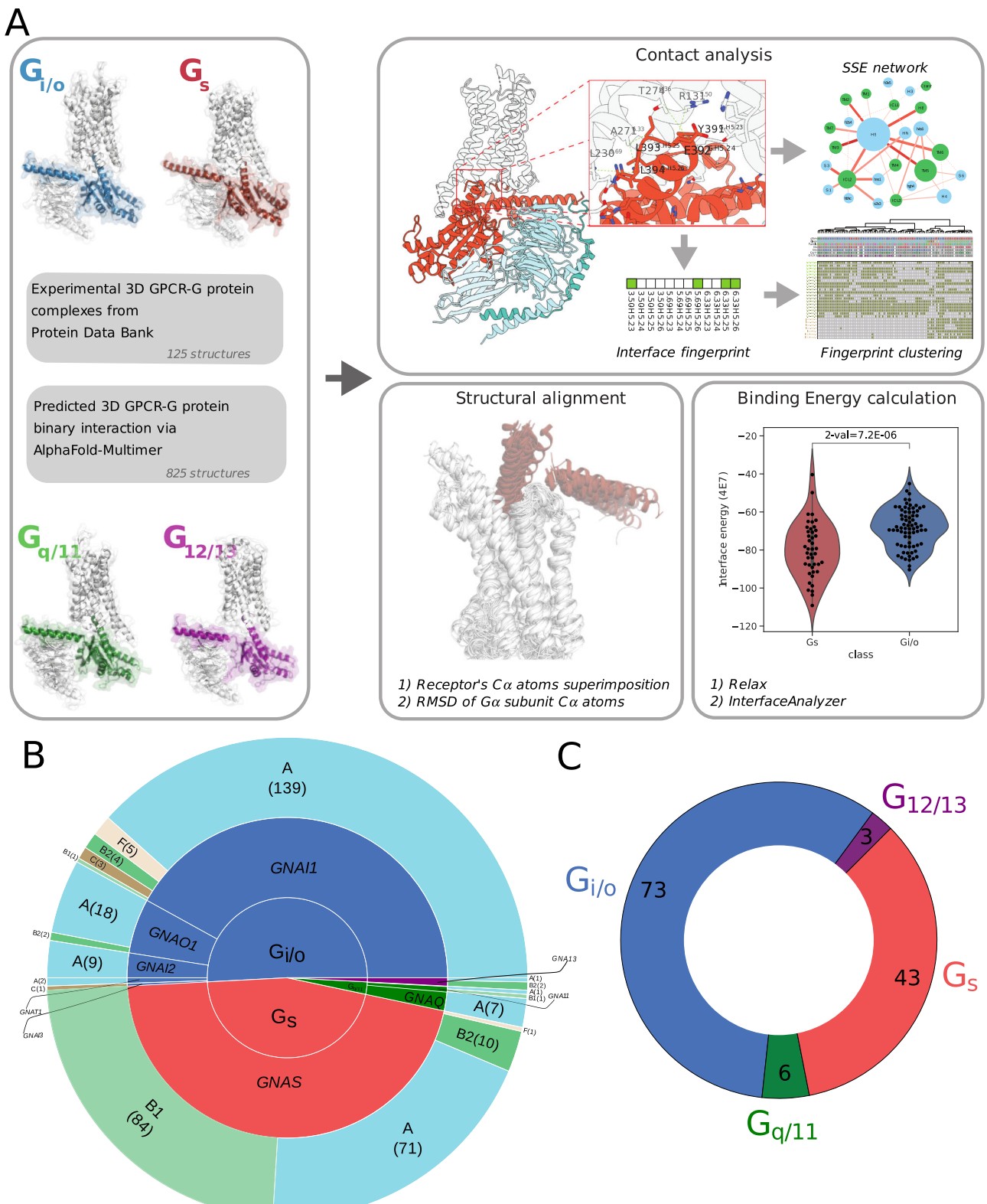

**Fig. 1 | workflow of the procedure and experimental structure statistics. A** workflow of the analysis procedure; **B** statistics of the total number of GPCR-G-protein complexes considered; **C** number of representative GPCR-G-protein used for downstream analysis.

hand, the following contacts are exclusively enriched in $G_{i/o}$ complexes: 2.39-G.H5.24, 3.49-G.H5.23, 3.50-G.H5.25, 3.53-G.H5.22 (or G.H5.23). Particularly striking is the enrichment of the contact involving the highly conserved $D^{3.49}$, which is found exclusively in $G_{i/o}$ complexes (Fig. 3 and Fig. 4E, G). Contacts mediated by the DRY $R^{3.50}$ or

3.53 positions, while enriched in $G_{i/o}$ complexes (Fig. 3), also mediate contacts in $G_s$ complexes (e.g. 3.50-G.H5.23, Fig. 4D, E). Other contacts specifically enriched in $G_{i/o}$ complexes are: ICL2.51-G.hns1.3, ICL2.51-G.s2s3.2, ICL2.54-G.S3.1, 5.71-G.H5.9, 6.25-G.h4s6.9, 6.28-G.h4s6.9, 6.29-G.H5.17, 6.29-G.H5.26, 6.33-G.H5.20 (G.H5.25) as well as

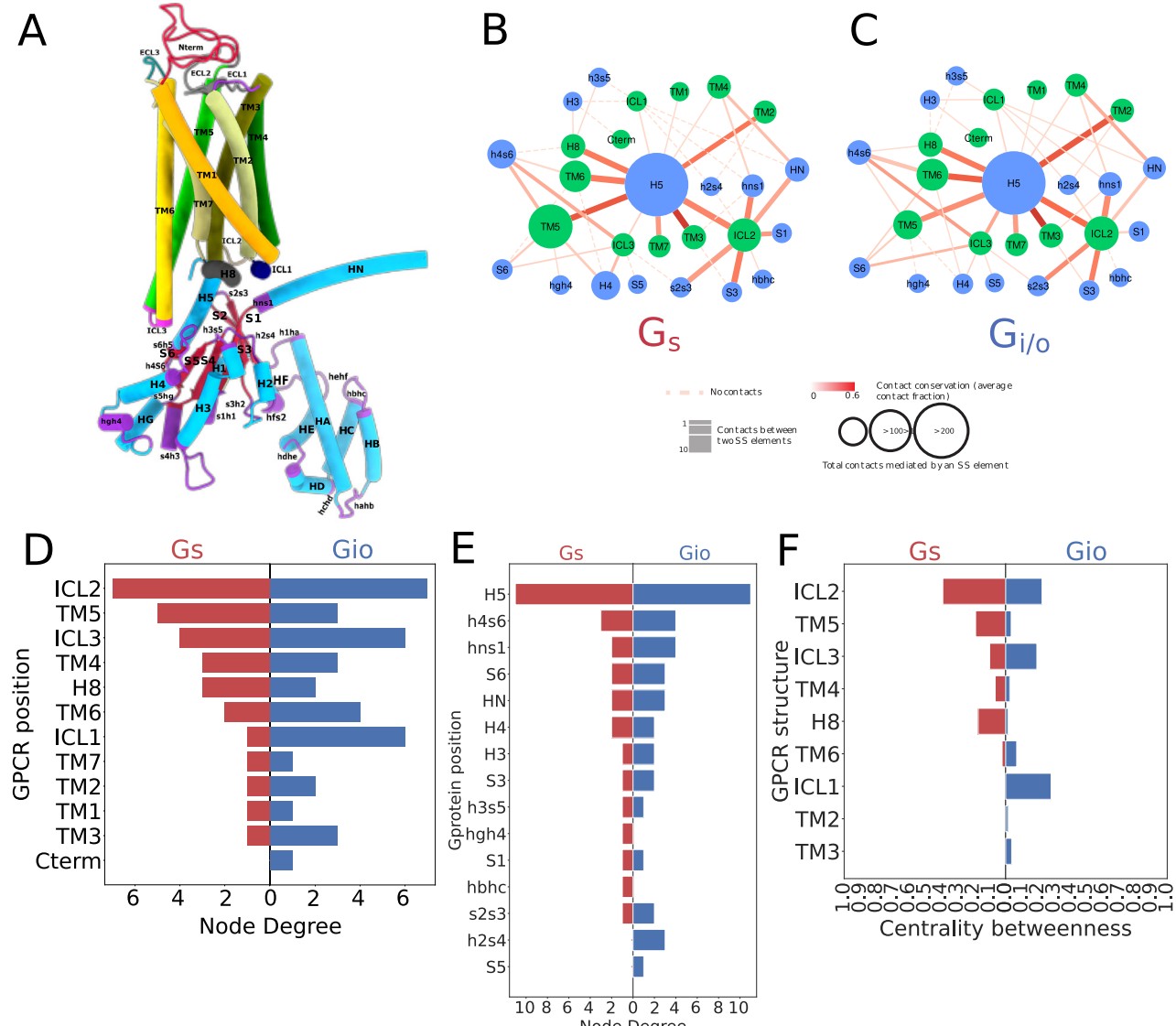

**Fig. 2 | Interface contact network analysis. A** A representative 3D complex structure (PDB 6CMO) with GPCR and G-protein SSE labels; **B** SSE contact network for $G_s$ complexes: GPCR and G-protein nodes are colored in green and cyan, respectively. Node diameter is proportional to the total number of contacts mediated by that SSE. Edge thickness is proportional to the number of contacts between connected SSEs and coloring (darker red) is directly proportional to contact conservation; **C** SSE contact network for $G_{i/o}$ complexes. Network characteristics as in 2 A; **D** GPCR SSE network node degree distribution for $G_s$ and $G_{i/o}$ networks; **E** G-protein SSE network node degree distribution for $G_s$ and $G_{i/o}$ networks; **F** GPCR SSE network betweenness centrality distribution. Source data are provided as a Source Data file.

all the contacts mediated by positions 8.47, 8.48, 8.49, 8.50 (Figs. 3A and 4E).

Overall, GPCR positions such as 5.61, 5.64, 5.65, 5.69, 5.76, 6.39, and 6.40 are enriched in $G_s$ complexes (Fig. 4D), while position 2.37, 2.39, 3.49, ICL2.50-2.55, 6.25, 6.29, 6.33, 6.34, 6.37, 7.53, or 8.50 are enriched in $G_{i/o}$ complexes (Fig. 4E). Likewise, the G-protein contact positions specifically enriched in $G_s$ complexes are h4s6.20, h4s6.3, H4.26, H5.11 (Fig. 4D), while positions h4s6.9, h4s6.12, s6.1, H5.9, H5.21, H5.22, H5.26 are enriched in $G_{i/o}$ complexes (Fig. 4E).

Notably, certain GPCR positions hold switch characteristics, in other words, some of the contacts that they mediate are enriched in $G_s$ and others in $G_{i/o}$ depending on the partner residues. For example, the contact of 5.65 with G.H5.16 is enriched in $G_{i/o}$, while the ones with G.H5.25 and G.H5.26 are in $G_s$. Similar patterns are observed for distinct contacts mediated by positions 5.68, 5.69 and 5.72 (Fig. 3 and Fig. 4D, E).

## Switching G-protein selectivity through contact interface mutation

To demonstrate the effect on G-protein coupling of the identified contact fingerprints, we employed a multistate-design computational protocol[35] to design receptor mutants specific for a selected G-protein and with reduced affinity for the others, which we then validated through the NanoBit G-protein dissociation assay[14] (Fig. 5A; Methods). We chose as starting templates for the design the structures of CCKAR, which has been solved in complex with both $G_s$ (PDB ID: 7EZK) and $G_{i/o}$ (PDB ID: 7EZH), and we focused the mutagenesis on the receptor positions forming the contact pairs most enriched in $G_s$ or $G_{i/o}$ complexes (Supplementary Data 4). We carried out two sets of designs: in one hand we sought to retain $G_s$ while removing $G_{i/o}$ couplings, and on the other, we maintained $G_{i/o}$ while reducing binding to $G_s$ (Fig. 5A). We found that certain mutations were more recurrent in top-designed sequences (Fig. 5A; see Methods). In particular, mutations A303[6.25]K,

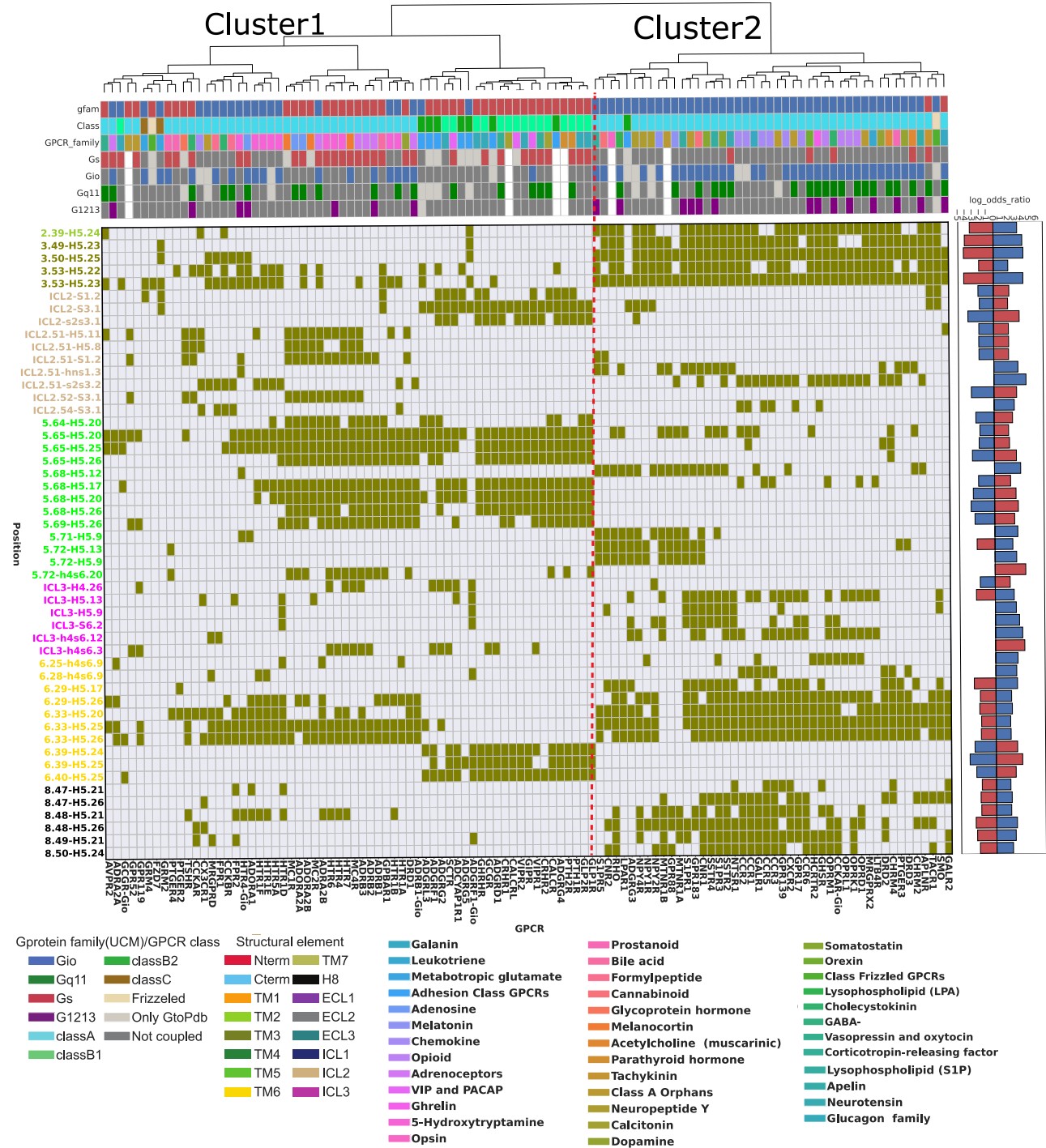

**Fig. 3 | Fingerprint of the GPCR-G protein interface.** GPCR-G-protein contact interface fingerprint (or CF fingerprint): each row is a GPCR-G-protein contact position (referenced respectively to GPCRdb numbering and G-protein position (CGN) numbering) and each column is a unique receptor complex. If a receptor is complexed with more than one G-protein, its complex fingerprint is reported accordingly. Columns are color-annotated to indicate: G-protein bound in the experimental structure, GPCR class, and experimentally reported coupling (according to UCM, or either GEMTA, Shedding, or GtoPDB). The right-side plot indicates the log-odds ratios (LORs) of the contacts observed at each position. Only contacts present in at least 10% of the structures and having an absolute LOR value greater than 2 are considered. Source data are provided as a Source Data file.

V311$^{6.33}$H, K375$^{8.48}$R and R376$^{8.49}$L were predicted to reduce G$_{i/o}$ while retaining G$_s$ binding, whereas mutations S149$^{ICL2.53}$A, V151$^{ICL2.55}$K, K308$^{6.30}$R, and K375$^{8.48}$P were recurrently predicted to reduce G$_s$ while retaining G$_{i/o}$ binding (Fig. 5A–D). These were subsequently tested through the NanoBiT G-protein dissociation assay using the NanoBiT-G$_s$ and NanoBiT-G$_{i1}$ sensors. Among the eight designed mutations, two

(V311$^{6.33}$H and R376$^{8.49}$V) and one (K308$^{6.30}$R) were found to be G$_s$-over-G$_i$ biased and G$_i$-over-G$_s$ biased, respectively, and the rest of the five had no effect on the G$_s$-vs-G$_i$ balance (Fig. 5E−G and Supplementary Fig. 4). We note that expression level of V311$^{6.33}$H was equivalent to that of WT (1:4). These data confirmed the importance of the identified contacts in switching coupling preferences between these two G-proteins.

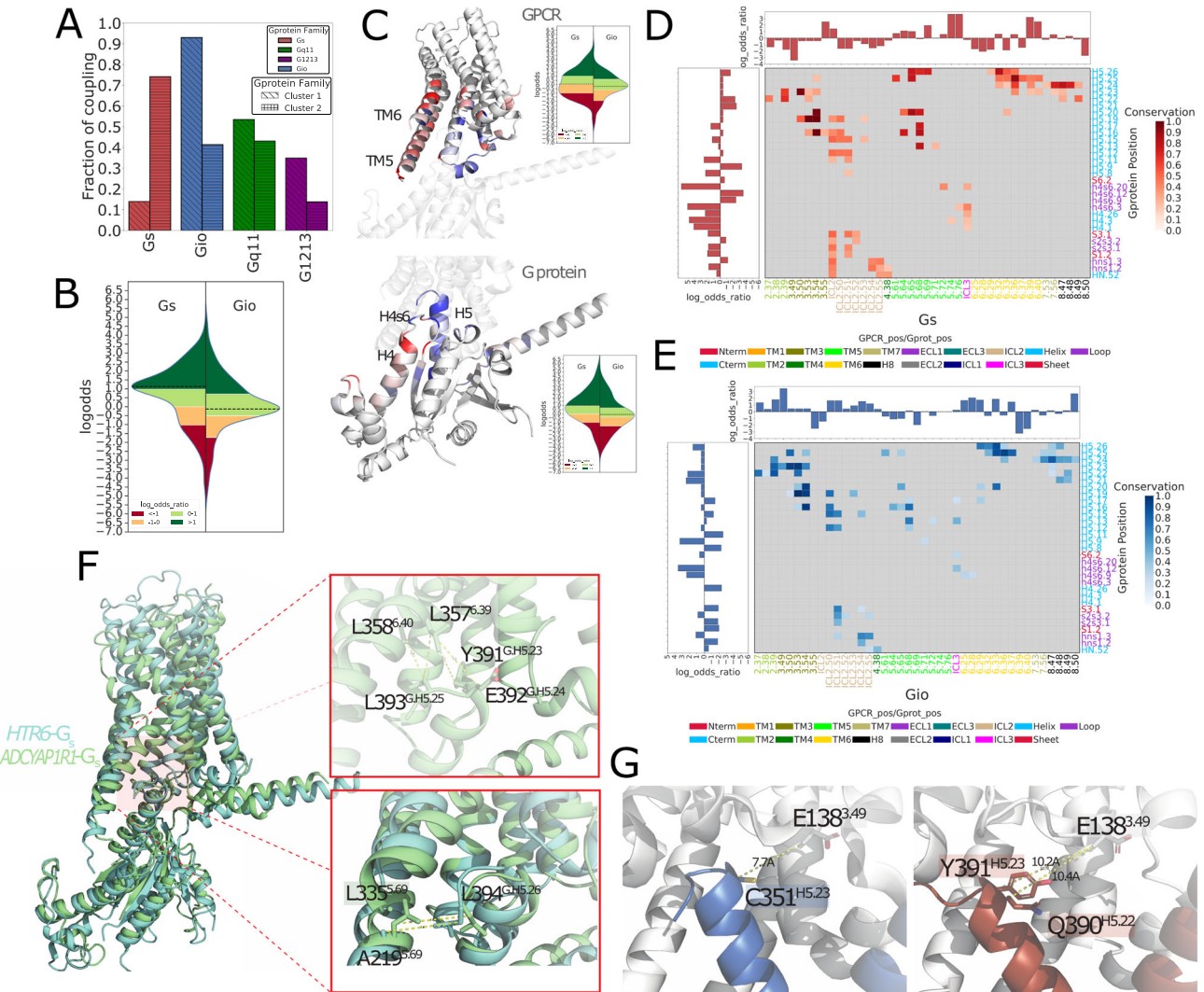

**Fig. 4 | Analysis of the GPCR-G protein contact fingerprints. A** fractions of experimental coupling groups of the receptors clustered identified through CF clustering; **B** violin plot showing the distribution of the LOR statistics for GPCR-G-protein contacts. Dashed lines indicate the median value; **C** $G_{i/o}$ log-odds ratio statistics represented with a color scale ranging from blue (negative LOR) to red (positive LOR) using B-factor annotations on a representative structure (PDB ID: 7VL9): receptor (top, chain R), G-protein (bottom, chain A) along with distribution of the LOR statistics for GPCR (top) and G-protein (bottom) contact positions. Dashed lines in the violin plots indicate the median value; **D** $G_s$ complexes contact frequency heatmaps: columns are GPCR positions in GPCRdb numbers, and rows are G-protein positions in CGN numbers. Only contacts with frequency > 20% (over the number of unique complexes) are considered; **E** Gi/o complexes contact frequency heatmaps: representation features are as in 4D; **F** structural comparison of $G_s$ complexes mediated by a class A (HTR6; cyan; PDB: 7XTB) and class B representative (ADCYAP1R1; light green; PDB 6P9Y) and zoomed view of the contacts mediated by GPCR positions 6.39, 6.40 and 5.69, respectively with G-protein positions H5.23, H5.24, H5.25, H5.26; **G** zoomed view of the contacts mediated by GPCR position 3.49 in a representative $G_{i/o}$ complex (CCKAR; PDB 7EZH) and distances between closest G-protein amino acids to E138 3.49 in the $G_s$ complex of the same receptor (PDB: 7MBX). Source data are provided as a Source Data file.

## Different repertoires of G proteins docking modes

We assessed the overall 3D similarity of GPCR-G-protein complexes via structural alignment, with a particular focus on the docking mode similarity of the G-protein α-subunits with respect to the receptor. To this end, we first superimposed the Cα-atoms of the most conserved positions within the 7TM bundle (i.e. that are present in all the solved structures) and then calculated the Root Mean Squared Deviation (RMSD) of the Cα-atoms of conserved positions of the Gα subunit (Fig. 1A; see Methods). The clustering of 3D complexes based on their RMSD shows that $G_s$ complexes tend to cluster separately from $G_{i/o}$ ones (Pmann-whitney = 2.5E-14; P-permanova = 1E-6; Fig. 6, Fig. 7A and Supplementary Fig. 5). The largest cluster comprises only Class A receptors, the vast majority bound to $G_{i/o}$ proteins, with the only exception of a few $G_s$ complexes (i.e. MC2R, GALR2), as well as the BDKRB2-GNAQ and S1PR2-GNA13 complexes (Fig. 6). The second largest cluster involves the vast majority of $G_s$ complexes, including Class

A and the totality of class B receptors. This cluster also comprises several $G_{q/11}$ (i.e. CCKAR, CCKBR, CRHR2, HRH1), $G_{i/o}$ (i.e. OPRK1, OPRD1, OPRL1, OPRM1, CX3CR1, ADGRF1 and ADGRG3) as well as two $G_{12/13}$ complexes (i.e. ADGRG1 and ADGRL3). Finally, a third, smaller outgroup cluster contains complexes involving class A, C, and F receptors most deviating from the other structures. Also in this case, the receptors in the largest, $G_{i/o}$ enriched cluster show a more promiscuous tendency, with $G_{q/11}$ and $G_{12/13}$ as most recurrent secondary couplings and $G_s$ as the least recurrent one (Fig. 6 and Fig. 7A). When considering only class A receptors, the RMSD distributions are no longer significantly different (Pmann-whitney = 6.7E-1; Supplementary Fig. 6), although characterized by significantly different centroids (P-permanova = 1E-6). We have also estimated residue level deviations of the Gα subunits of the fitted complexes by calculating Root Mean Square Fluctuations (RMSF; see Methods) and compared the profiles obtained for $G_s$ and $G_{i/o}$ complexes, which highlighted significantly

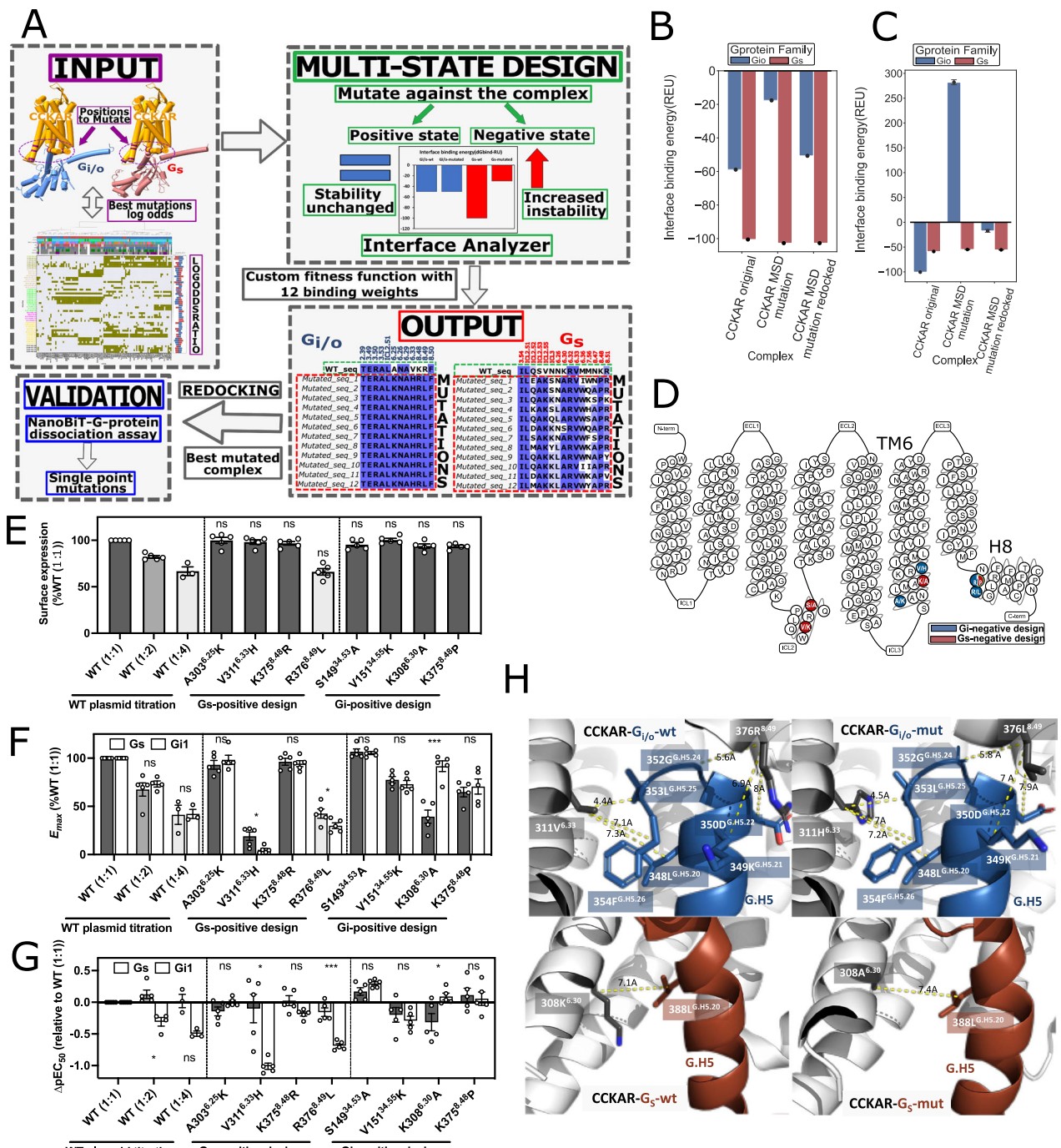

higher fluctuations for $G_{i/o}$ complexes with respect to $G_s$ ones (Wilcoxon $P = 1.18E-13$), for the whole Ras GTPase domain and in particular for regions such as H2, H3 as well as the C-terminal lobe of the Ras domain (Fig. 7C). Overall, $G_s$ complexes display less variability of the terminals, with H5 appearing more conformationally restrained and bent towards TM3 and ICL2, while $G_{i/o}$ complexes display greater conformational variability for αN and H5 (Fig. 7D left). A comparison of representative structures of the four coupling groups show slight differences in the docking mode of each representative, which are nevertheless smaller for $G_{i/o}$, $G_q$, and $G_{12/13}$ (Fig. 7E).

We also explored the potential conformational bias of the nanobodies (i.e. Fab16 or Nb35) used to stabilize the bound G-protein on the observed G-protein docking modes. First, we annotated the presence/absence of the nanobody for each complex subjected to RMSD clustering. We observed no correlation between RMSD clusters and the

presence or absence of nanobodies (Fig. 6 and Supplementary Fig. 5). Second, we relaxed the GPCR-heterotrimeric G-protein complex without such nanobodies, using state-of-the-art methods for structural refinement (Rosetta relax; see Methods). Both RMSD and RMSF analysis performed on relaxed structures showed even larger statistically significant differences between $G_s$ and $G_{i/o}$ complexes (Pmann-whitney = 1.1-36; Supplementary Fig. 7). Notably, the differences of the $G_s$ and $G_{i/o}$ RMSD distributions is still significant when considering only class A receptors (Pmann-whitney = 6.5E-6; Supplementary Fig. 8).

## Different energies characterize specific GPCR-G-protein interfaces

We exploited the relaxed GPCR-heterotrimeric G-protein complexes to further characterize the binding interface energy of the complex using Rosetta InterfaceAnalyzer[36] (see Methods). By considering all available

**Fig. 5 | Multistate design of coupling-switch mutants and in-vitro validation.**
**A** Multi-State design workflow; **B** Comparison of binding interface energy estimation (REU, InterfaceAnalyzer) of mutated and redocked structures in $G_{i/o}$ negative multi-state design, $n = 12$ independent experiments. Data represent mean ± SEM for CCKAR MSD mutation ($G_{i/o}$ −17.625 ± 0.00, $G_s$ −102.715 ± 0.00) and CCKAR MSD mutation redocked ($G_{i/o}$ −50.688 ± 0.00, $G_s$ −102.715 ± 0.00); **C** Comparison of binding interface energy estimation (REU,InterfaceAnalyzer) of mutated and redocked structures in $G_s$ negative multi-state design, $n = 12$ independent experiments. Data represent mean ± SEM for CCKAR MSD mutation ($G_{i/o}$ −54.912 ± 0.31, $G_s$ 282.023 ± 2.54) and CCKAR MSD mutation redocked ($G_{i/o}$ −55.5983 ± 0.31, $G_s$ −17.3738 ± 1.51); **D** Snake plot visualization of CCKAR structure from GPCRdb (https://gpcrdb.org/protein/cckar_human/) annotated with mutation tested in experimental validation (blue: $G_{i/o}$ negative design; red: $G_s$ negative design); **E** Cell-surface expression of the CCKAR mutants. HEK293 cells transiently expressing the indicated CCKAR construct along with the NanoBiT-$G_{i1}$ sensor were subjected to the flow cytometry analysis. WT (1:2) and WT (1:4) denote 2-fold and 4-fold less volumes, respectively, of transfected plasmids than the mutant plasmids. Data are from 3–5 independent experiments with each dot representing an individual experiment. Error bars represent SEM. Statistical analysis was performed using the two-way ANOVA with Sidak's correction for multiple comparisons (A303$^{6.25}$K, V311$^{6.33}$H, K375$^{8.48}$R, S149$^{34.53}$A, V161$^{34.55}$K, K308$^{6.33}$A, K375P with WT (1:1); R376$^{8.49}$L with WT (1:4). ns, $P > 0.05$). $n = 5$, except for WT (1:4) ($n = 3$). **F, G** $G_s$- and $G_i$-coupling activity of the CCKAR mutants. HEK293 cells transiently expressing the indicated CCKAR construct along with the NanoBiT-$G_s$ or the NanoBiT-$G_{i1}$ sensor were subjected to the NanoBiT-G-protein-dissociation assay using CCK-8 as a ligand. The ligand-response parameters $E_{max}$ (**F**) and $\Delta pEC_{50}$ (**G**), which were normalized to WT (1:1), were used to denote the G-protein-coupling activity. Data are from 3–5 independent experiments with each dot representing an individual experiment. Error bars represent SEM. Statistical analysis was performed using the multiple paired, two-sided t-test between $G_s$ and $G_{i1}$ (ns, $P > 0.05$; * $P < 0.05$; *** $P < 0.001$). $n = 5$, except for WT (1:4) ($n = 3$) and WT (1:2) in $G_{i1}$ ($n = 4$); **H** Representation of contacts of the successful mutations for $G_{i/o}$-negative design (upper panel; V311$^{6.33}$H and R376$^{8.49}$L) for wild-type CCKAR (PDBID:7EZH, upper-left) and mutated CCKAR (upper-right) and Gs-negative design (lower panel; K308$^{6.30}$A) for wild type CCKAR (PDBID: 7EZH, lower left) and mutated CCKAR (lower-right).

GPCR Gα-subunit pairs with an experimentally resolved complex, we showed that the ΔG of binding of $G_s$ complexes is significantly lower than $G_{i/o}$ complexes (Pmann-whitney = 7.2E-6; Fig. 8A) and it partially correlates with the slightly higher ΔSASA observed for $G_s$ complexes compared to $G_{i/o}$ ones (Pmann-whitney = 1.4E-3; Fig. 8B). When considering class A receptors only, the difference in binding energy distribution between $G_s$ and $G_{i/o}$ complexes is even larger (Pmann-whitney = 3.4E-6; Supplementary Fig. 9). Intriguingly, we observed that $G_s$ is bound less strongly to class B1 than class A receptors (Pmann-whitney = 2.3E-3; Fig. 8C), suggesting that receptors from different classes might bind to the same G-protein with different affinities due to different structural and functional requirements. On the other hand, the same receptor always binds with higher affinity to $G_s$ than $G_{i/o}$. Indeed, we have compared the binding energies of complexes of the same receptor (e.g. GCGR, CCKAR, and HTR4) with both $G_s$ and $G_{i/o}$ proteins. Notably, the ΔG of binding for $G_s$ is always lower and is characterized by higher ΔSASA compared to $G_{i/o}$ irrespective of the slight docking modes variations observed in $G_s$ complex structures of the same receptor (Fig. 8D–F).

## AlphaFold2 predictions extend our understanding of the structural basis of coupling diversity

To further our understanding of the structural basis of GPCR-G-protein recognition, we predicted through AlphaFold-multimer (v2.3)[37] 996 GPCR · G-protein alpha subunit pairs from UCM binary interactions (see Methods). We first benchmarked AF-multimer by assessing the prediction performances with or without 3D templates for 125 representative GPCR-G-protein complexes with available experimental structures. Using 3D templates in the prediction slightly improves the performances, assessed by measuring the deviation of predicted vs. experimental interfaces via either the DockQ metric (Wilcoxon P = 0.021; Fig. 9A) or the fraction of native contacts (Wilcoxon P = 0.004; Fig. 9B). It must be emphasized that even without templates multimer's predictions achieve almost comparable performances, yielding an average DockQ score of 0.645 vs. 0.664 obtained for predictions with templates (Fig. 9A). We, therefore, ran multimer predictions with available templates for receptor-heterotrimeric G-protein complexes whose receptor-α subunit binary interactions are reported in the UCM dataset. Since some predicted complexes showed unrealistic docking topology, we created a composite filter based on structural, topological, and statistical scoring metrics (e.g. pDockQ) to remove predicted structures with unrealistic complex topologies as well as regions predicted with low confidence (based on pLDDT; see Methods). Such a filtering scheme improves the correlation between prediction scores (pDockQ) and distance from experimental structures (DockQ) in the benchmark (Fig. 9C, D). We applied this filter to

the multimer's predictions, yielding a set of 825 complexes for downstream processing (Supplementary Data 5). Contact analysis performed on predicted complexes revealed patterns similar to those observed experimentally, in particular, the models recapitulated most of the experimental contacts of $G_s$ and $G_{i/o}$ complexes (Fig. 9E). As expected, the agreement of contact patterns from experimental and predicted $G_{q/11}$ and $G_{12/13}$ complexes is lower due to fewer structural templates available for this coupling groups (Fig. 9E). Notably, contact heatmaps derived from each G-protein group display highly specific patterns, which could potentially illuminate signaling mechanisms for poorly studied G-proteins such as $G_{12/13}$. For instance, contact frequency heatmaps for $G_{12/13}$ complexes display peculiar patterns mediated by TM5, ICL3, and TM6 with G.H5. Several of the contacts observed for other G-protein complexes at positions G.H5.25 and G.H5.26 are missing (Fig. 9E). On the other hand, contacts mediated by positions G.H5.23 and G.H5.24 with 6.33, as well as 6.29, appear as highly specific for $G_{12/13}$. We also confirmed on predicted structures the differences in binding energies observed between $G_s$ and $G_{i/o}$ complexes (Pmann-whitney = 9.8E-4; Fig. 9F). We also found that $G_{q/11}$ complexes are as stable as $G_s$, while $G_{12/13}$ ones are less affine, similarly to $G_{i/o}$ (Pmann-whitney = 9.8E-4; Fig. 9F). Such differences in binding energies are anti-correlated with differences in Interface Area (Fig. 9H). Additionally, we found interclass significant differences for $G_s$PCRs, with ClassA being the most and ClassC the least stable (Pmann-whitney = 5.8E-3; Fig. 9H).

## Discussion

In the present study, we have performed a computational, comparative analysis of 3D GPCR-G-protein complexes in their nucleotide-free state, to identify structural hallmarks of the interaction interfaces that might be linked to coupling specificity.

Complexes involving different G-proteins are characterized by distinctive structural signatures, like contact networks displaying a different engagement of secondary structural elements such as TM5, TM6, and ICLs. This notion is in line with earlier comparative analysis highlighting a selectivity filter operated by TM5 and TM6 (commented on[28]). More recently, structural determination of serotonin receptors in complex with either $G_s$ or $G_i$ proteins, accompanied by bioinformatics analysis of a representative set of class A complexes, supported the role of these secondary structure elements by highlighting a macro-switch, operated by TM5 and TM6 terminals' variable length, which dictates selectivity towards $G_s$ vs $G_{i/o}$[30]. Our unsupervised analysis of interface contacts entails complementary interactions between key positions on both receptor's and G-protein's sides. $G_s$ complexes are characterized by significantly higher fractions of enriched contacts, that are mainly imposed by contacting positions on the

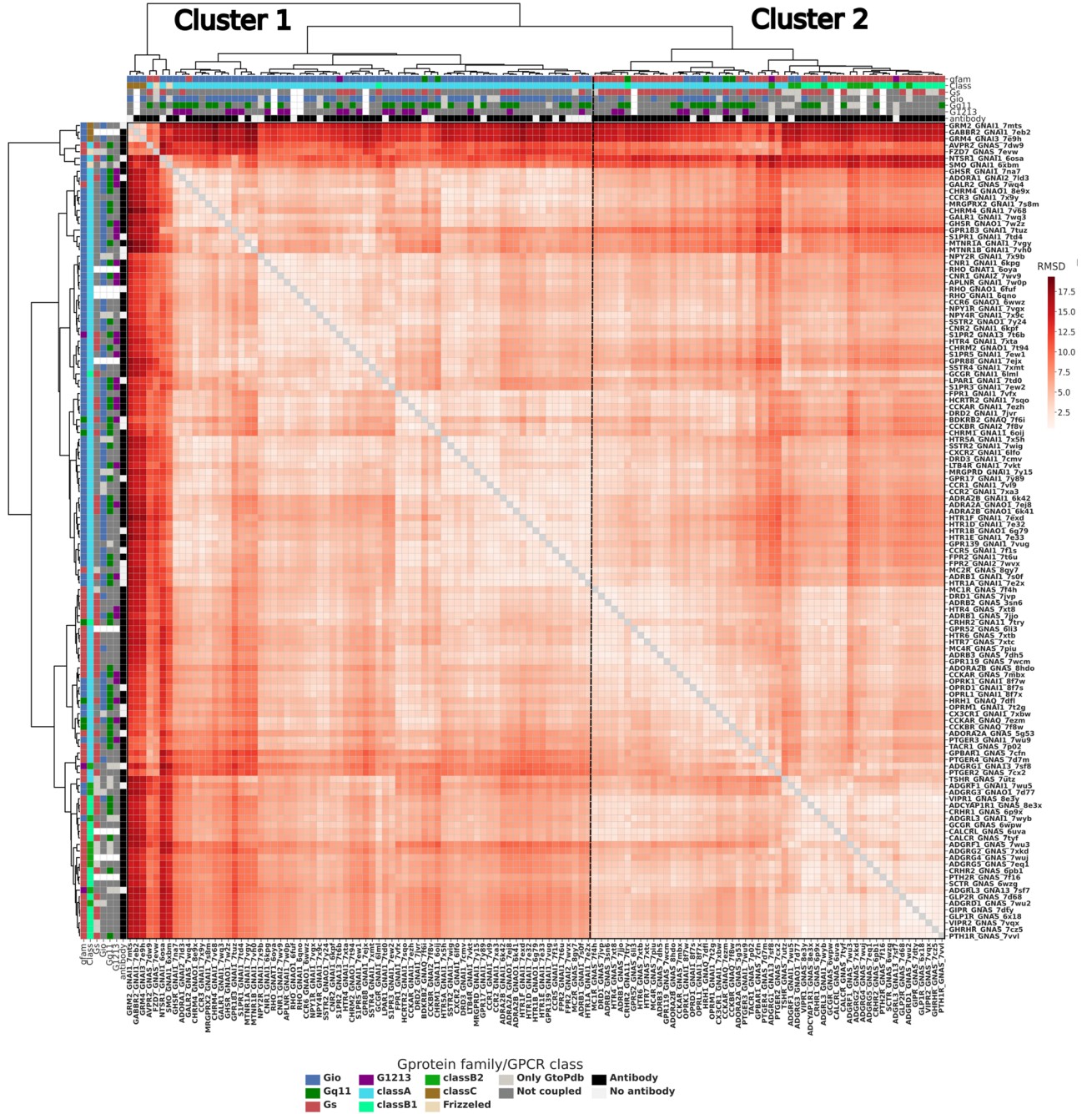

**Fig. 6 | Clustering GPCR-G protein complexes based on docking mode similarity: RMSD hierarchical clustering heatmap.** Color annotations are as in Fig. 3 with an additional last field containing the information regarding presence of nanobodies. Source data are provided as a Source Data file.

G-protein's side. However, the combination of GPCR and G-protein residues leads to maximum discrimination between the $G_s$ and $G_{i/o}$ groups. These results are reminiscent of the G-protein barcode model for coupling specificity, which emerges by the presentation of an evolutionary more rigid G-protein barcode to a more flexible receptor counterpart[20]. At the same time, $G_s$ bound receptors are characterized by a less promiscuous binding, suggesting that the structural requirements for $G_s$ specific binding overall impose discrimination for secondary couplings.

Certain SSEs are exclusively characterized by contact enrichments for specific G-protein (e.g. TM2, TM3 or H8 for $G_{i/o}$). Notably, the DRY motif-mediated contacts, particularly $D^{3.49}$'s ones, can be considered one of the main structural hallmarks of $G_{i/o}$ vs $G_s$ complexes. The latter is indeed characterized by a greater bending of G-protein's H5 towards

TM3's C-term and ICL2, which function as hinges to concomitantly detach the G-protein C-term from the DRY motif. Other regions are characterized by a switch-like character, meaning that certain positions form contacts enriched in $G_{i/o}$ and others in $G_s$, including: ICL2, which encompasses the known ICL2.51 specificity determinant position[38], TM5, ICL3 and TM6. We noted that the switching behavior might pertain also to individual positions and depend on the specific contact partner. These residue-level features likely work in coordination with the macro-switch described for the serotonin receptors[30]. We leveraged our findings to implement a computational multi-state design procedure to predict mutations biasing a given coupling, starting from a promiscuous WT sequence (e.g. CCKAR), which we experimentally verified via NanoBiT G-protein dissociation assays. Experiments identified two mutations (V311$^{6.33}$H and R376$^{8.49}$V) to bias

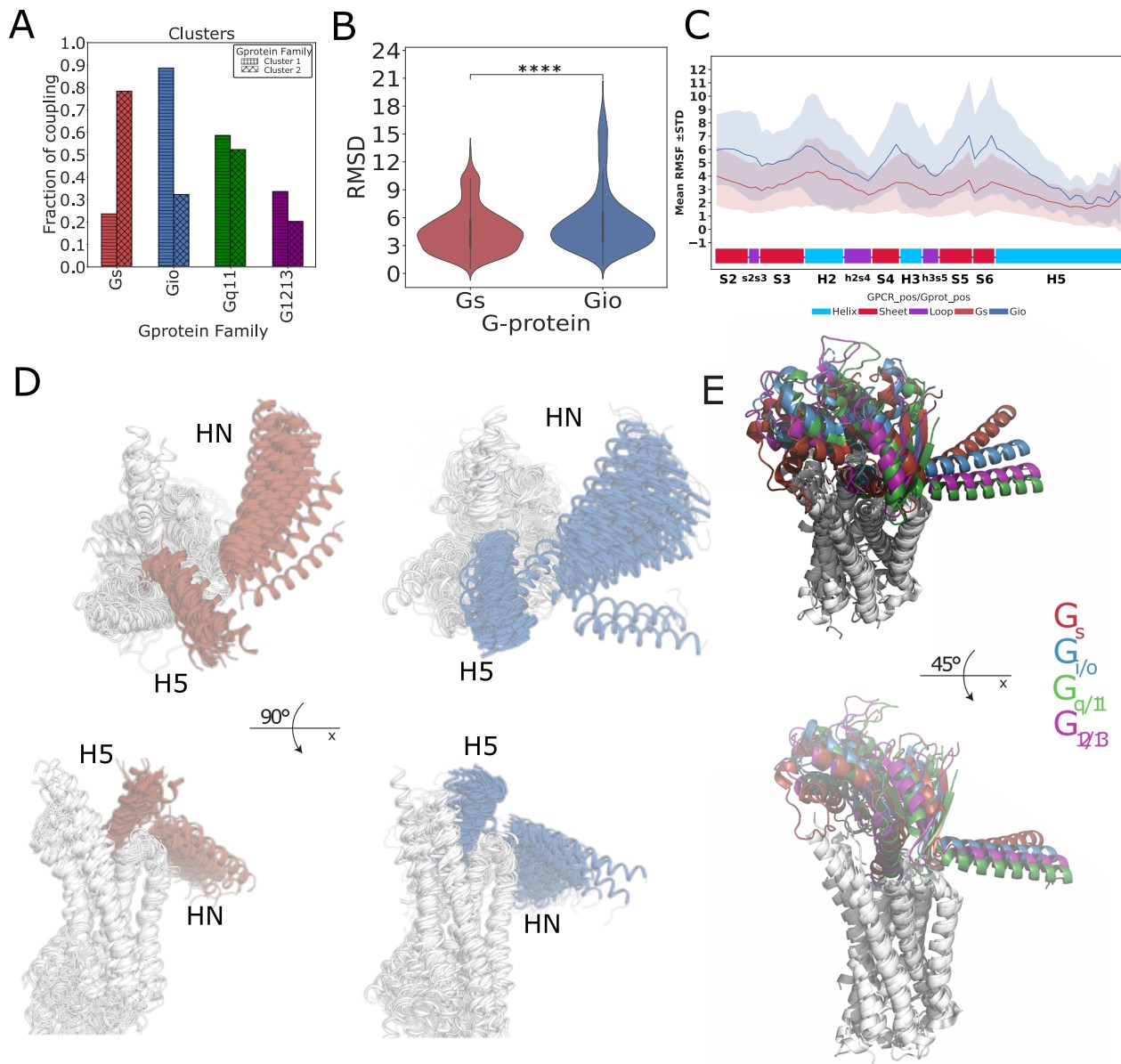

**Fig. 7 | Measuring similarity of G-protein docking modes. A** Barplot of the coupling preferences of receptors of Cluster 1 and 2; **B** distribution of the RMSD within $G_s$ and $G_{i/o}$ complexes, $n = 2628$ experimental structure pairs for $G_{i/o}$ and $n = 903$ experimental structure pairs for $G_s$, p-value has been computed with the Wicoxon rank-sum, two-sided test with Bonferroni correction $P = 2.512E\text{-}14$ (**** $P < 0.0001$). Boxplots show the median as the centre and first and third quartiles as bounds of the box; the whiskers extend to the last data point within 1.5 times the interquartile range (IQR) from the box's boundaries; **C** root mean squared fluctuations of the G-protein consensus positions, each point is represented as mean ± SD calculated from $n = 125$ experimental structures. P has been computed via a Wilcoxon rank-sum test with Bonferroni correction ($P = 1.18E\text{-}13$); **D** superposition of class $G_s$ and $G_{i/o}$ representative complexes: GPCR 7TM bundles are represented as white cartoons; the N-term and C-term of the $G_s$ (red, left) and $G_i$ (blue, right) alpha subunits are represented as marker of the G-protein structural variability on experimental complexes; **E** structural superimposition of representative structures defined on the basis on minimum RMSD to other members of the group (for $G_s$ and $G_{i/o}$) and release date ($G_{q/11}$): $G_s$ (PDB: 7XTB; red), $G_{i/o}$ (PDB: 7VL9; blue), $G_{q/11}$ (PDB: 7EZM; green), $G_{12/13}$ (PDB: 7T6B;purple). Source data are provided as a Source Data file.

$G_s$ and one (K308[6.30]R) to bias $G_i$ signaling, confirming the importance of the identified contacts in switching coupling preferences between these two G-proteins.

Through our comprehensive analysis, we show that the few $G_{q/11}$ and $G_{13}$ complexes available display certain structural characteristics similar to $G_{i/o}$ complexes. These correlate with recent phylogenetic analysis showing that $G_i$ and $G_q$ family members share a common ancestor, and that $G_{12/13}$'s ancestor is likely a retro-gene originated by retroposition from a pre-$G_q$ gene[39].

The usage of a state-of-art AI model (i.e. AlphaFold-multimer[37]) for structural prediction also allowed us to expand the structural

repertoire of GPCR-G-protein complexes. This is particularly valuable for poorly characterized groups, such as $G_{12/13}$ ones. Indeed, we predicted peculiar contact patterns at the TM5 and ICL3 that are characteristic of this group and might suggest unique structural requirements. Indeed, the critical importance of these regions also emerged in our previous effort to engineer a $G_{12}$-DREADD, which was achieved by swapping shorter ICL3 loops from GPR183 or GPR132 on hM3D and experimentally validated to be functional[14].

The observed structural differences are linked with different predicted binding affinities for the distinct coupling group complexes,

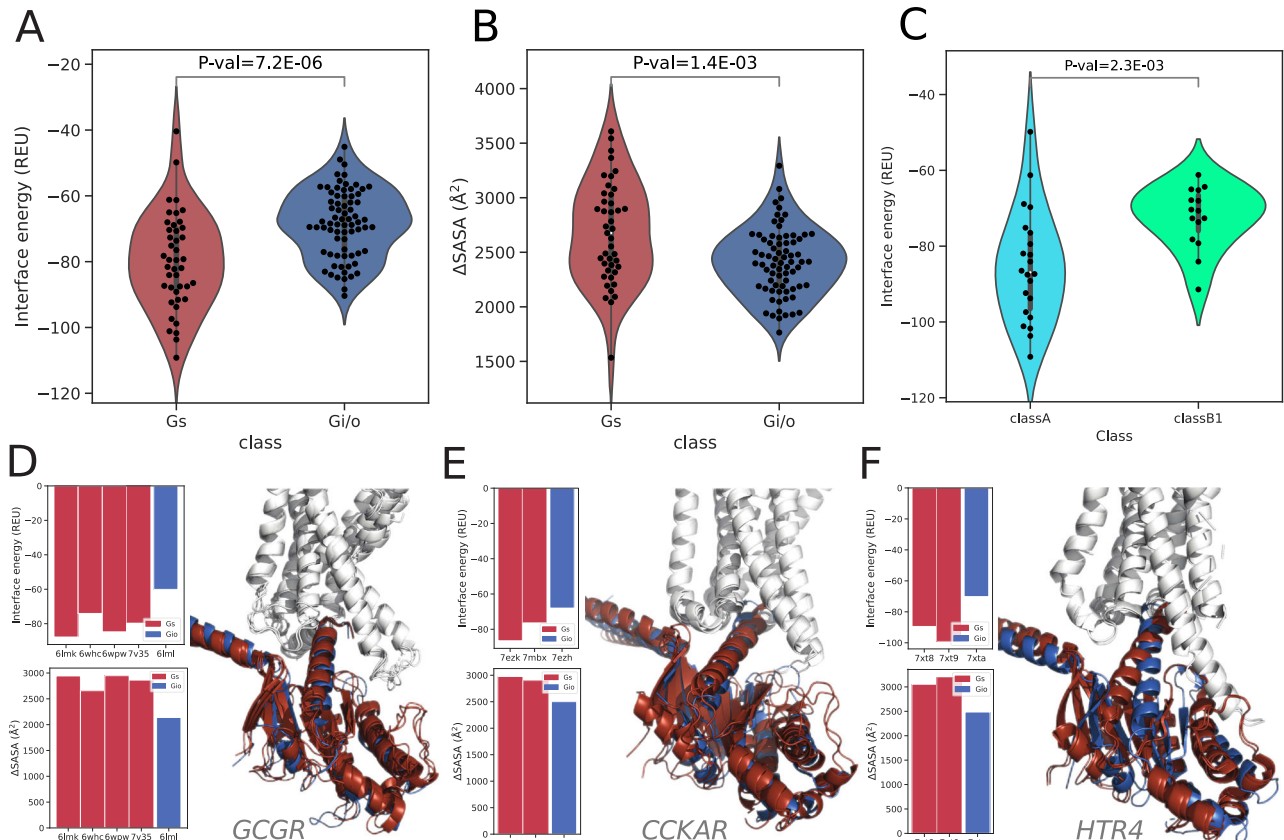

**Fig. 8 | Binding energy estimated through Rosetta. A** ΔG binding (REU); **B** Delta Solvent Accessible Surface Area (ΔSASA); $n = 43$ structural complexes for $G_s$ and $n = 73$ structural complexes for $G_{i/o}$; **C** ΔG binding (REU) of $G_s$ complexes with classA and classB receptors, n = 22 complexes for Class A GPCR and n = 15 receptors for Class B1 GPCR. Interface energy, ΔSASA and superimposed 3D cartoon models of $G_s$ (red) and $G_{i/o}$ (blue) complexes of **D** GCGR; **E** CCKAR; **F** HTR4. The p-values have been computed with two-sided Mann–Whitney U test. Boxplots show the median as the centre and first and third quartiles as the bounds of the box; the whiskers extend to the last data point within 1.5 times the interquartile range (IQR) from the box's boundaries. Source data are provided as a Source Data file.

with $G_s$ complexes being more stable than $G_{i/o}$ ones. Slight differences in interface contacts and docking mode might reflect the difference in binding affinities for $G_s$ observed for class A vs. class B receptors. We speculate that the higher binding energy of class B receptor, suggestive of less stable binding to $G_s$, might be related to slower rates for G-protein activation observed for a representative class B receptor (GCGR) compared to class A one (ADRB2)[34]. On the other hand, a comparison of the binding energies of $G_s$ and $G_{i/o}$ complexes also revealed significant differences, which holds true even when considering the complexes with the same receptor, further stressing the major contribution from the G-protein residues in priming the binding. Such differences in binding energies are also confirmed in AF-multimer predicted models, which also allow us to characterize the members of other families such as $G_{q/11}$ and $G_{12/13}$. We speculate that $G_s$ has a higher energetic barrier for activation due to its peculiar biological role of AC activation, which should be tightly regulated. The conformational restriction in $G_s$ complexes might contribute to spatio-temporally fine-tune AC activation. On the other hand, looser $G_{i/o}$ binding to cognate receptors could explain their reported faster nucleotide turnover[40]. Moreover, lower structural conservation and less affine $G_{i/o}$ complexes are likely connected to the success of this coupling[14,33], which is instrumental in providing a redundant mechanism for AC inhibition. Structural features and energetics are certainly not the only factors governing the evolutionary success of a certain coupling: indeed $G_{12/13}$, which form complexes predicted as weak as the $G_{i/o}$ ones, are the least successful couplings. Higher-order spatio-temporal dynamics, such as the more recent evolution of $G_{12/13}$[39], might explain these patterns.

A limitation of this study is that most of the structures considered are in the nucleotide-free state. Indeed, previous studies have suggested that intermediate states other than the nucleotide-free complexes influence G-protein-coupling selectivity[41]. Nucleotide-decoupled G proteins mutants, which bypass the intermediate-state complexes, are characterized by a degradation of coupling selectivity, indirectly highlighting the importance of the conformational dynamics of these intermediate-state complexes in guiding selectivity[42]. Future structural studies systematically targeting the intermediate states of GPCR-G protein complexes will improve the understanding of coupling selectivity.

The greater availability of experimental complex structures and increasingly accurate predicted models, including context-aware and diffusion-based deep learning techniques to predict alternative conformational states, will allow us to better understand the structural basis of G-protein-coupling specificity in the future. This knowledge will be key to design better-biased drugs, able to modulate only certain transducers, as well as it will be leveraged to improve the design of novel chemogenetic probes such as DREADDs.

## Methods

### Data sets

We used the mapping between PDB[43] and Pfam[44] provided by SIFTS[45] (2023/01/28 update) to retrieve all the structures containing both a GPCR and a G-protein α subunit. We used the Pfam entry PF00503 to identify structures of G-α subunits, and Pfam entries PF00001 (rhodopsin receptor family - class A), PF00002 (secretin receptor family - class B), PF00003 (class C receptors), PF01534 (Frizzled/Smoothened

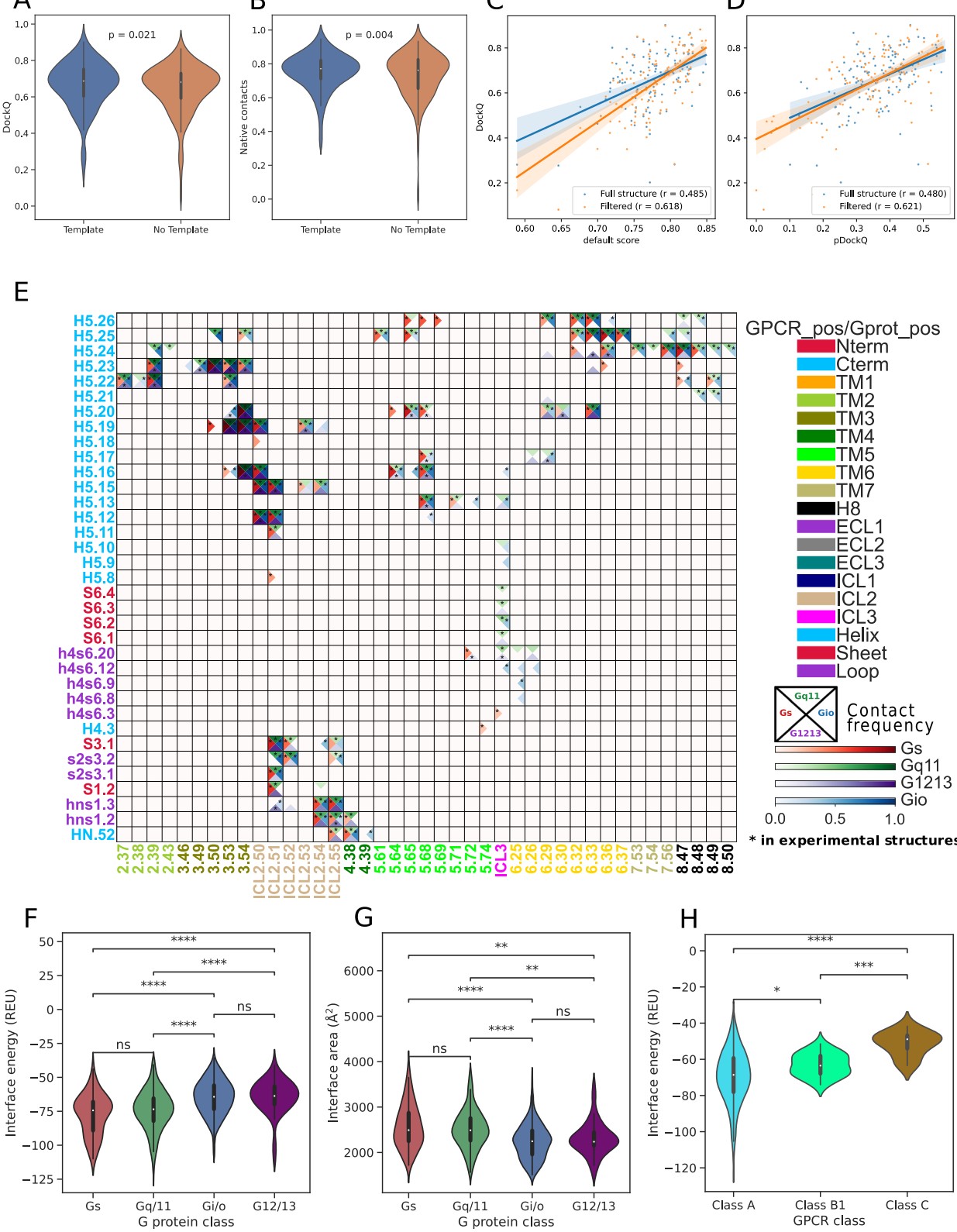

family) to identify GPCRs. We found 376 structures that met this criterion. If more than one GPCR or Gα chain were in the structure, we considered the pair of chains with the highest number of contacts between them (Supplementary Data 1).

We considered as contacts all the pairs of amino acids with less than 8 Å between their Cβ (Cα for glycine - see below), following standard practices employed for contact analysis in structural

predictions[46]. Only the structures in which all the contacts between the GPCR and Gα chains were mapped to the same pair of Uniprot accessions (according to SIFTS xml residue level mappings) were kept for further analysis. We excluded the remaining structures that were considered as chimeric. In this way, we obtained 362 structures.

Whenever we found more than one structure representing the same GPCR-G-protein pair, we considered the one with the highest

**Fig. 9 | AlphaFold-multimer prediction of experimental GPCR-G protein complexes.** Evaluation of AlphaFold models with a resolved experimental structure. **A** DockQ and **B** fraction of retrieved native contacts for the models generated with and without the usage of available 3D experimental templates of the monomer structures. N = 125 structural complex models computed for both conditions. The p-values have been computed with a two-sided Wilcoxon signed-rank test. Scatterplot of the DockQ of AlphaFold models computed using structural templates as a function of **C** the default score and **D** the pDockQ. Translucent bars represent 95% confidence intervals estimated with bootstrap. Filtered models contain only the residues with pLDDT ≥ 70. Binding energy of the filtered AlphaFold models estimated through Rosetta; **E** Alphafold complexes contact frequency heatmap for $G_s$ (red scale), $G_{i/o}$ (blue scale), $G_{q/11}$ (green scale) and $G_{12/13}$ (purple scale), each corresponding to a differently colored triangle: columns are GPCR positions in GPCRdb numbers, rows are G-protein positions in CGN numbers. Only contacts with frequency > 20% (over the number of unique complexes) are considered. The color intensity is proportional to the contact frequency. Experimental contacts are marked with a black asterisk; for E) all GPCR classes and F) only class A receptors. **F** ΔG binding (REU) predicted for AlphaFold-multimer complexes; p-values: Gs v.s. Gq/11 = 0.67; Gq/11 v.s. Gi/o = 1.55e-15; Gi/o v.s. G12/13 = 1; Gs v.s. Gi/o = 3.28e-11; Gq/11 v.s. G12/13 = 1.30e-5; Gs v.s. G12/13 = 4.06e-6; **G** Delta Solvent Accessible Surface Area (ΔSASA); *p*-values: Gs v.s. Gq/11 = 1; Gq/11 v.s. Gi/o = 7.34e-16; Gi/o v.s. G12/13 = 1; Gs v.s. Gi/o = 5.52e-8; Gq/11 v.s. G12/13 = 3.45e-3; Gs v.s. G12/13 = 8.06e-3. The analysis considers $n$ = 87 Gs, $n$ = 265 Gq/11, n = 430 Gi/o, and n = 43 G12/13 structural complex models; **H** ΔG binding (REU) of complexes with class A, class B1, and class C receptors. The *p*-values have been computed with a two-sided Mann–Whitney U test with Bonferroni correction. ns $P > 0.05$; * $P < 0.05$; ** $P < 0.01$; *** $P < 0.001$; **** $P < 0.0001$. Boxplots show the median as the center and the first and third quartiles as the bounds of the box; the whiskers extend to the last data point within 1.5 times the interquartile range (IQR) from the box's boundaries. Source data are provided as a Source Data file.

resolution to be the representative for each complex. In case that more than one structure had the same resolution, we chose the structure which covered the largest portion of the corresponding G-protein according to SIFTS. We found 125 GPCR-Gprotein pairs with at least one solved structure (Supplementary Data 1).

### GPCR-Gα complexes prediction via AlphaFold-Multimer

We used Alphafold-Multimer v2.3.13732 to generate 3D models of GPCR-heterotrimeric G-protein complexes. The human canonical protein sequences of GNB1 and GNG1 were used as the β and γ subunits, respectively. We trimmed the N-terminal part of the GPCRs up to 50 residues before the start of TM1, to avoid folding the long extracellular portion of some receptors. The databases required to run AlphaFold-Multimer were downloaded on 12 January 2023. Among the 5 models generated for each GPCR-G-protein complex, only the best one was considered for further analysis. The score used to evaluate the models was the default one used by AlphaFold-Multimer (0.2*pTM + 0.8*ipTM).

To assess the reliability of AF-multimer models, we predicted the structure of the 125 GPCR-G-protein complexes with available experimental structures. We used the human sequences of the corresponding GPCR-Gα pairs in all predictions. We generated 3D models "using templates", that is allowing AlphaFold-Multimer to use the 3D structural models available in the PDB (they are used only to predict the structure of single chains), and "without templates", running the predictions setting --max_template_date=01-01-1900 instead, to avoid the usage of any available experimental template.

A total of 996 GPCR-Gα pairs, reported to bind in UCM[33], were considered, respectively corresponding to 164 and 13 human GPCRs and Gα proteins (the three members of the transducin family were not considered). We removed 21 predicted complexes (975 structures left) based on unrealistic complex topologies, i.e. not having any of the following G-protein's H5 positions, i.e. H5.16, H5.19, H5.20, H5.23, H5.24, H5.25, that we found to most recurrently mediate contacts with GPCR's residues in experimental structures. To further remove low-quality models produced by Alphafold, we used the pDockQ metric[47], which is fine-tuned on predicting the DockQ of the predicted complex with respect to experimental complexes. We kept only the structures with pDockQ ≥ 0.23[48] (118 out of 975 structures removed)[36].

We then employed Alphafold's confidence score (predicted Local Distance Difference test - pLDDT), to remove protein terminals predicted with low confidence which might lead to artefactual contacts. Hence, we removed all the residues with pLDDT <70. We then performed the interface analysis on these trimmed sequences using the same procedure used for experimental complexes. As a last filter to remove low-quality structures, we used the output of Rosetta InterfaceAnalyzer[36] (https://www.rosettacommons.org/docs/latest/ application_documentation/analysis/interfacE-analyzer; see below paragraph "Analysis of GPCR-Gα binding energy with Rosetta") and we kept only models with ΔSASA ≥ 1500 Å², which is slightly less than the minimum ΔSASA found in all experimental structures. In this way we removed 32 more structures, leading to a total of 825 high-quality complex structures for downstream analysis (Supplementary Data 5).

### Contact analysis

We considered residue-residue contacts mediating GPCR-G-protein interfaces as those having the Cβ spatially closer than 8 Å (Cα for glycine) (as in[46]). We analyzed 362 solved PDB GPCRs-G-protein complexes, which according to G protein family classification comprised 184 $G_{i/o}$, 166 $G_s$, 9 $G_{q/11}$, and 3 $G_{12/13}$ structures. We mapped interface PDB residues to Uniprot canonical sequences residues by using SIFTS residue level mappings from individual PDB XML files. The interface sequence positions were then mapped to GPCRdb numbering[31] (Supplementary Data 2) and to the Common G-protein Numbering (CGN) schemes[32] (Supplementary Data 3). We aggregated contacts of different structures referred to the same GPCR-G-protein complex and considered equivalent residues pairs from different structures only once to avoid redundancy. We compared $G_{i/o}$ and $G_s$ complexes by creating a consensus list of GPCR and G-protein positions found in contact with at least one of the members of the two groups. For each GPCR-G-protein consensus positions, we calculated the fraction of GPCRs displaying a contact in each G-protein family group (either $G_{i/o}$ or $G_s$) over the total number of GPCRs in contact. Such a fraction reflects the conservation of contact in each G-protein group. We constructed an interface contact network by considering the consensus list of GPCR-G-protein contacts from all complexes. Contacting positions were projected to the secondary structure elements of both GPCRs and G-proteins, represented as nodes of the networks. Node diameter is proportional to the total number of contacts mediated by the position of that secondary structure element. Edge width is proportional to the number of unique GPCRs mediating contact between the two linked SSEs. Edge coloring (bright to dark red) is proportional to the average of the contact fraction of individual position pairs, and it reflects the overall contact conservation between two SSEs. Dashed lines indicate contact not formed in each G-protein class but present in the other. We calculated network statistics such as node degree and centrality betweenness distribution through Cytoscape (https://cytoscape.org/)[49] and customized Python scripts. All the analyzes have been done through customized Python scripts, using biopython libraries (version 1.78). Network drawings have been generated through Cytoscape.

To analyze the statistical significance of the difference between $G_s$ and $G_{i/o}$ contact fingerprints, we generated an inter-chain contact graph for each structure. We then used the Frobenius norm of the difference between the adjacency matrices of the interchain contact

**Table 1 | Contingency table for calculating log-odds ratio**

| Contact pair/G-protein | Contact | No contact |
|---|---|---|
| Coupled | CC | CN |
| Not coupled | NC | NN |

graphs as a distance metric for the structures. The resulting distance matrix was used to perform a PERMANOVA test[50] to determine if the graphs generated by interaction with the $G_s$ and the $G_{i/o}$ family were significantly different from each other. This test compares the difference in docking modes in the same G-protein family to the difference in docking modes between different G-protein families. To evaluate the difference in variance between the distribution of intra-family pairwise distances, we used the PERMDISP test. Both tests were performed using the scikit-bio library (version 0.5.4) in python (https://cran.r-project.org/web/packages/vegan/index.html). We performed $10^6$ and $10^4$ permutations for PERMANOVA and PERMDISP, respectively.

Analysis was also done on Alphafold predicted complexes considering 430 $G_{i/o}$, 265 $G_{q/11}$, 87 $G_s$ and 43 $G_{12/13}$ structures according to UCM. Networks have been created for all position pairs as well as for cases where >0.2 complexes have the position pairs.

### Fingerprint analysis

We have generated interface fingerprints by creating position vectors by mapping the contacts (1 if present, 0 otherwise) of either residue pairs (Complex fingerprints, or CF), or the individual positions separately for the receptor and G-protein (Receptor and G-protein fingerprints, respectively RF and GF). We performed unsupervised, hierarchical clustering on rows (unique GPCR-G-protein complex) using the "*Ward*" method and Euclidean distance as metric, employing the *clustermap* function from *seaborn* library (version 0.11.1). Rows are color annotated to indicate: G-protein bound in the experimental structure, GPCR class, experimentally reported couplings (according to UCM). The top plot indicates the enrichment of the contacts observed at each position.

For each consensus position of GPCR and Gprotein we calculated the log-odds ratio (LOR) from contingency Table 1 using the following equation:

$$LOR = \log\left(\frac{CC}{NN} \times \frac{NC}{CN}\right) \tag{1}$$

*CC* and *CN* terms represent the number of GPCRs coupled to a specific Gprotein group ($G_{i/o}$ or $G_s$) that are or are not, respectively, in contact at that position (either individual GPCR or G-protein positions or residue pairs). *NC* and *NN* terms represent the number of non-coupled GPCRs for a specific G-protein, that are or are not in contact, respectively, at a given position (either individual GPCR or Gprotein positions or residue pairs). Contacts contributed from the loops, N-termini and C-termini of the GPCR where aggregated. We calculated the binning statistics of the log-odds ratio of contacts.

### Rosetta multistate design predictions of CCKAR coupling switching mutations

To bias $G_s$ over $G_{i/o}$ coupling, or vice versa, we have designed mutations through the Multi-state design by Rosetta[35,51], as availale in the RosettaCommon software suite (version 2021.16.61629). As a starting template for the design, we have chosen CCKAR (UniProt Accession: P32238), a cholecystokinin receptor that has been solved in complex with $G_s$ (PDB ID: 7EZK), $G_{i/o}$ (PDB ID: 7EZH") and $G_{q/11}$ (PDB ID: 7EZM)[52]. The CCKAR positions selected for design were either involved in specific contacts, based on comparative contacts analysis of $G_s$ (7EZK) and $G_{i/o}$ (7EZH) experimental complexes, or those with the

highest log-odds ratios from GPCRomE-wide contact pair statistics. To switch $G_s$ coupling, we chose to mutate the following positions: 3.54, ICL2.51, ICL2.52, ICL2.53, ICL2.55, ICL3(299), 6.26, 6.30, 6.32, 6.33, 6.36, 7.56, 8.47, 8.48, and 8.51. To switch $G_{i/o}$ coupling we chose the following positions: 2.39, 3.49, 3.50, 3.53, ICL2.51, 6.25, 6.26, 6.29, 6.33, 8.48, 8.49, and 8.50. Next, we designated positive state and negative state complexes (7ezk or 7ezh). The positive state being the one whose structure and binding energy are preserved by mutations and the negative state being the one which is destabilized by mutations. We used a custom fitness function to describe the energy states of binding interfaces of both complexes with 12 different binding weights (1-12) and getting for each of them the best fitted structure with a different sequence of mutations for destabilizing either $G_s$ or $G_{i/o}$ coupling. We calculated the binding energy of the interface with InterfaceAnalyzer (https://www.rosettacommons.org/docs/latest/application_documentation/analysis/interfacE-analyzer) and redocked the structures with RosettaDock using local refinement of the interface between GPCR and G-protein structures[53]. We chose the best redocked structures based on the Total score of the redocking tool and calculated the interface binding energy (REU) with InterfaceAnalyzer.

To redock the mutated GPCR-G-protein complexes we used the Rosetta docking protocol with a docking:docking_local_refine flag to refine GPCR-Gα interface. We created top 20 redocked structures for each of the complexes created by MultiState design binding weights (1–12) and chose the best redocked structure for each of them based on the Total score provided by the protocol[53]. Then we ran Rosetta InterfaceAnalyzer for the best redocked structures and assess the binding interface energy.

### NanoBiT- G- protein-dissociation assay for designed CCKAR mutants

Ligand-induced G-protein dissociation was measured by the NanoBiT-G-protein dissociation assay[14], in which the interaction between a Gα subunit and a Gβγ subunit was monitored by the NanoBiT system (Promega). Specifically, a NanoBiT-G-protein consisting of the Gα subunit fused with a large fragment (LgBiT) at the α-helical domain (Gα-LgBiT) and an N-terminally small fragment (SmBiT)-fused $Gγ_2$ subunit with a C68S mutation (SmBiT-$Gγ_2$-CS) was expressed along with untagged $Gβ_1$ subunit and a test CCKAR construct. The full-length human CCKAR was inserted into the pCAGGS expression vector with an N-terminal fusion of the hemagglutinin-derived signal sequence (ssHA), FLAG epitope tag and a flexible linker (MKTIIALSYIFCLVFA-DYKDDDDKGGSGGGGSGGSSSGGG; the FLAG epitope tag is underlined). The resulting construct was named as ssHA-FLAG-CCKAR. HEK293A cells (Thermo Fisher Scientific, cat no. R70507) were seeded in a 6-well culture plate at a concentration of $2 \times 10^5$ cells ml-1 (2 ml per well in DMEM (Nissui) supplemented with 5% fetal bovine serum (Gibco), glutamine, penicillin and streptomycin), one day before transfection. Transfection solution was prepared by combining 5 μL (per dish hereafter) of polyethylenimine (PEI) Max solution (1 mg ml⁻¹; Polysciences), 200 μL of Opti-MEM (Thermo Fisher Scientific) and a plasmid mixture consisting of 200 ng ssHA-FLAG-CCKAR (or an empty plasmid for mock transfection), 100 ng Gα-LgBiT subunit ($Gα_s$-LgBiT or $Gα_{i1}$-LgBiT), 500 ng $Gβ_1$ subunit and 500 ng SmBiT-$Gγ_2$-CS subunit. For $Gα_s$-LgBiT, to enhance expression of the NanoBiT-$G_s$ sensor, 100 ng RIC8B plasmid was co-transfected. After incubation for 1 day, the transfected cells were harvested with 0.5 mM EDTA-containing Dulbecco's PBS, centrifuged and suspended in 2 ml of HBSS containing 0.01 % bovine serum albumin (BSA; fatty acid-free grade; SERVA) and 5 mM HEPES (pH 7.4) (assay buffer). The cell suspension was dispensed in a white 96-well plate at a volume of 80 μL per well and loaded with 20 μL of 50 μM coelenterazine (Angene) diluted in the assay buffer. After a 2 h incubation at room temperature, the plate was measured for baseline luminescence (SpectraMax L, Molecular Devices) and titrated

concentrations of sulfated CCK-octapeitide (Peptide Institute, cat no. 4100-v; 20 μL; 6X of final concentrations) were manually added. The plate was immediately read for the second measurement as a kinetics mode and luminescence counts recorded from 5 min to 10 min after compound addition were averaged and normalized to the initial counts. The fold-change values were further normalized to those of vesicle-treated samples and used to plot the G-protein dissociation response. Using the Prism 9 software (GraphPad Prism), the G-protein dissociation signals were fitted to a four-parameter sigmoidal concentration-response curve with a constrain of the HillSlope to absolute values less than 2. For each replicate experiment, the parameter Span (= Top – Bottom) of the individual CCKAR mutants were normalized to those of WT CCKAR performed in parallel and the resulting Emax values were used to calculate ligand response activity of the mutants.

## Flow cytometry

Plasmid transfection for the ssHA-FLAG-CCKAR and the NanoBiT-Gi sensor was performed according to the same procedure as described in the "NanoBiT-G-protein-dissociation assay". One day after transfection, the cells were collected by adding 200 µl of 0.53 mM EDTA-containing Dulbecco's PBS (D-PBS), followed by 200 µl of 5 mM HEPES (pH 7.4)-containing Hank's Balanced Salt Solution (HBSS). The cell suspension was transferred to a 96-well V-bottom plate in duplicate and fluorescently labeled with an anti-FLAG epitope (DYKDDDDK) tag monoclonal antibody (Clone 1E6, FujiFilm Wako Pure Chemicals, cat no. 012-22384; 10 µg per ml diluted in 2% goat serum- and 2 mM EDTA-containing D-PBS (blocking buffer)) and a goat anti-mouse IgG secondary antibody conjugated with Alexa Fluor 488 (Thermo Fisher Scientific, cat no. A11001; 10 µg per ml diluted in the blocking buffer). After washing with D-PBS, the cells were resuspended in 200 µl of 2 mM EDTA-containing-D-PBS and filtered through a 40-µm filter. The fluorescent intensity of single cells was quantified by an EC800 flow cytometer equipped with a 488 nm laser (Sony). The fluorescent signal derived from Alexa Fluor 488 was recorded in an FL1 channel, and the flow cytometry data were analyzed with the FlowJo software (FlowJo). Live cells were gated with a forward scatter (FS-Peak-Lin) cutoff at the 390 setting, with a gain value of 1.7. Values of mean fluorescence intensity (MFI) from approximately 20,000 cells per sample were used for analysis. Typically, we obtained an MFI value of 2000 (arbitrary unit) for WT CCKAR and 20 for the mock transfection. For each experiment, we normalized an MFI value of the mutants by that of WT performed in parallel and denoted relative levels.

## Clustering of G-proteins complex conformations

We compared $G_{i/o}$ and $G_s$ complexes by performing Root Mean Square Deviation (RMSD)-based clustering. To calculate RMSD, we created a list of consensus positions based on all the sequences of GPCR and all G-protein in 362 complexes, by first mapping PDB residues to Uniprot canonical sequences via SIFTS[45] and then to GPCRdb consensus numbers[31]. We considered 141 GPCR and 73 G-protein consensus positions defining respectively the consensus core of the 7TM domain and the Ras GTPase domain solved in all experimental structures. The complexes were fitted using the Cα atoms of the GPCR core, while the Cα of the G-protein core were used to calculate the RMSD after superimposition. Calculations were performed using the Superimposer function of the PDB Biopython module[54] (version 1.78) through customized scripts. We performed hierarchical clustering on RMSD using the Ward method with Euclidean distance as metrics, using the *clustermap* function from *seaborn* library (version 0.11.1). We compared the distribution of the RMSD calculated among complexes of the $G_{i/o}$ and $G_s$ groups using a Wilcoxon rank-sum test. Results were displayed through matplotlib (https://matplotlib.org/) and seaborn

(https://seaborn.pydata.org/) libraries using customized python scripts. We also calculated the root mean squared fluctuations of the G-protein consensus positions using the following equation:

$$\rho_i = \sqrt{\left\langle (x_i - x_i')^2 \right\rangle} \qquad (2)$$

where $x_i$ is the coordinate of particle $i$ and $x_i'$ is the coordinate of particle $i$ in the reference structure ', which is the complex with the least RMSD deviation from the other complexes (i.e. centroid) in the $G_s$ (PDB: 8E3X) and $G_{i/o}$ (PDB: 8F7S) groups.

We compared $G_s$ and $G_{i/o}$ groups RMSFs by performing a Wilcoxon test and we plotted each position and its standard deviation.

## Analysis of GPCR-Gα subunit binding energy with Rosetta

To analyze the GPCR-Gα interface in a 3D structural model, we first relaxed the structure using the Rosetta relax application[55], using backbone constraints. Then we ran Rosetta InterfaceAnalyzer[36] (https://www.rosettacommons.org/docs/latest/application_documentation/analysis/interface-analyzer), from RosettaCommon software suite (version 2021.16.61629), specifying the chains of the GPCR and the Gα which are interacting in the complex. This protocol takes a multichain complex as input and computes a new structure in which the two chains of interest are separated. The interface energy and the ΔSASA are calculated as the difference in energy and SASA in the bound and unbound structure. We run InterfaceAnalyzer with the "-pack_input" and "-pack_separated" flags to optimize the side chain configuration before and after separating the chains. If a nanobody was present in a structure, we removed it before the relaxation step, to limit its influence on the analysis.

The interface energy is computed according to the Rosetta energy function, which includes physics-based terms that represent electrostatic and van der Waals' interactions, as well as statistical terms representing the probability of finding the torsion angles in the Ramachandran plots. This score is indicated in Rosetta Energy Units (REU) and cannot be converted into the actual binding energy, but it gives a reasonable estimation of the stability of the complex[56].

## Software

We employed Pymol (v2.4.1) and ChimeraX (v1.5) to generate 3D cartoon representations. We employed customized scripts in python (version 3.8.11), using matplotlib (v3.6.0), seaborn (v0.11.1), and biopython (v1.78) libraries. We calculated residue-residue contact by using a customized script derived from the CIFPARSE-OBJ C++ library (https://mmcif.wwpdb.org/docs/sw-examples/cpp/html/index.html).

## Reporting summary

Further information on research design is available in the Nature Portfolio Reporting Summary linked to this article.

# Data availability

Source data are provided with this paper. Data generated for this study are available at https://github.com/raimondilab/GPCR_structure_analysis and https://doi.org/10.5281/zenodo.8067369. GPCR-Heterotrimeric G protein complexes predicted with AF-multimer are also available via Precogx webserver (https://precogx.bioinfolab.sns.it/). The raw data used in this study are available in the Zenodo database under the accession code https://doi.org/10.5281/zenodo.8063796. The PDB accession codes used in this study can be found in Supplementary Data 1, corresponding to a total of 362 experimental structures analyzed. Source data are provided with this paper.

## Code availability

Code used for this study is available at https://github.com/raimondilab/GPCR_structure_analysis and https://doi.org/10.5281/zenodo.8067369.

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

## Acknowledgements

F.R. was supported by the Italian Ministry of University and Research through the Department of Excellence "Faculty of Sciences" of Scuola Normale Superiore. The research leading to these results also received funding from the Italian Association for Cancer Research (AIRC) under My First AIRC Grant (MFAG) 2020 - ID. 24317 project – P.I. Raimondi Francesco. A.I. was funded by KAKENHI JP21H04791, JP21H05113, and JPJSBP120213501 from the Japan Society for the Promotion of Science (JSPS); JP20gm0010004, JP20am0101095, JP22ama121038, and JP22zf0127007 from the Japan Agency for Medical Research and Development (AMED); JPMJFR215T and JPMJMS2023 from the Japan Science and Technology Agency (JST); Daiichi Sankyo Foundation of Life Science; Takeda Science Foundation; The Uehara Memorial Foundation. M.T. received JSPS KAKENHI 22J10475. We gratefully acknowledge the CINECA award, in collaboration with AIRC, for the availability of high-performance computing resources and support. We gratefully acknowledge the computational resources of the Center for High-Performance Computing (CHPC) at Scuola Normale Superiore. We thank Kayo Sato, Shigeko Nakano, and Ayumi Inoue at Tohoku University for their assistance in plasmid preparation and the cell-based GPCR assays. We also thank Tatsuya Ikuta for helpful discussion. We are grateful to Natalia De Oliveira Rosa for having uploaded the AF-predicted structures on Precogx (https://precogx.bioinfolab.sns.it/).

## Author contributions

M.M. and P.M. performed the computational analysis and wrote the manuscript; M.T. performed the in-vitro assays; A.I. designed the study and wrote the manuscript; F.R. designed the study, performed the computational analysis, and wrote the manuscript.

## Competing interests

The authors declare no competing interests.
