## [Peer Review File · Nature Communications]

GPCRome-wide analysis of G-protein-coupling diversity using a computational biology approachREVIEWER COMMENTS

Reviewer #1 (Remarks to the Author):

In their manuscript „GPCRome-wide structural analysis of G-protein-coupling selectivity” the authors report an extensive computational study that is intended to shed light on the still largely unresolved question of G protein-coupling selectivity. In brief, the authors use all currently available 3D GPCR/G-protein complex structures and AlphaFold2 predictions as a data basis to analyze residue-residue contacts between receptors and G proteins. These contacts are used in fingerprinting and clustering analyses to determine structural differences between coupling of different transducers to receptors. Moreover, the authors perform calculations of binding energies that suggest that Gs and Gi binding to receptors can be distinguished energetically, with Gs proteins forming more stable ternary complexes than Gi proteins.

This is an interesting albeit highly technical manuscript that, for non-specialists in the field and likely for many experimentalists, can be sometimes very hard to follow. There are three major issues with the manuscript that do not justify publication in Nature communications:

1) To me as an experimentalist, the conceptual novelty of this manuscript over previously published work is not clear. There have been numerous papers published in the past on this topic (e.g. PMID: 28489817; PMID: 31855179; PMID: 33846507; PMID: 35302494) and I cannot easily judge the advancement of this work over previous papers. It would be important to the reader to integrate all previous knowledge into the concept of the present manuscript.

2) Another significant drawback of the study is the complete lack of any experimental validation of the authors' results. First, it is very intriguing that the authors can devise potentially unique residues/contacts that are exclusively responsible for coupling to either Gi or Gs proteins. To substantiate these remarkable conclusions, it is important to e.g. design receptor or G-protein mutants at those residues and to demonstrate the presence or absence of an effect on G-protein coupling. Second, by using a promiscuous receptor, mutation of those residues specific for Gi/Gs-coupling should allow for the design of novel DREADDs. This should be demonstrated. Third, the peculiar contact patterns for G12/13-coupling should also be explored experimentally to e.g. unravel novel receptor/G12/13 pairings. Without any experimental validation, I do not think that this manuscript provides a sufficient conceptual advance in the field.

3) The computational exploration of receptor/G protein-binding energies is certainly important and interesting. However, it would be important to have some kind of experimental validation, e.g. ITC measurements or similar. Given the larger contact interface of receptors with Gs, it feels intuitive that the binding affinities are higher than those of Gi-complexes. Please confirm experimentally. On another note, the distribution of ΔG for Gs appears to be much wider than for Gi. This would mean that there are GPCR/Gs complexes with a lower free binding energy than Gi complexes. Please comment.

Reviewer #2 (Remarks to the Author):

This paper presents a useful analysis of all currently available structures of complexes of GPCRs with Galpha proteins. Such structural bioinformatics studies are important because they synthesize a great deal of data, and information scattered about 100s of scientific papers that would not be easily searched or collated by individual scientists. On the whole, I think Nature Communications should consider such papers for publication. This paper is well done and worth considering for publication.

The authors do a solid job of constructing their data set and presenting their statistical analysis of residue-residue interactions between GPCRs and Galpha proteins and how these differ between different Galpha families. They also examine binding energies of Galpha/GPCR complexes and how these differ by Galpha and GPCR classes. They also undertake the important task of using AlphaFold2 to predict the structures of all likely GPCR/Galpha complexes.

I have several suggestions for the authors to consider that might improve or extend their analysis, mostly to do with how the AlphaFold2 structures are utilized. The AF2 structures are an exciting part of this paper, potentially significantly extending our understanding of the whole GPCR/Galpha-ome, and they are given relatively short shrift in the paper. Unless the authors think they are not accurate enough, this section could be extended.

1) It is not clear if the AlphaFold2 predicted structures will be made publicly available. I think this should be a condition of publication. The coordinate files should be made available on some repository such as zenodo or the authors' github repository, and should be annotated properly in mmCIF format including sequence records and UniProt numbering of the coordinates (in "auth_seq_id"). If the authors download some models from the EBI AlphaFold2 site (<https://www.alphafold.ebi.ac.uk/>), they can see what records are there and mimic those.

2) It would make sense to present and analyze AF2 predictions of GPCR/Galpha structures that are already in the PDB and compare the AF2 predictions with the experimental structures. For this purpose, AF2 can be run without templates. AF2-multimer does not use heteromer templates anyway, but the results will be more applicable to structures without templates if the sequence query pairs with templates are also run with AF2 without PDB templates. The easiest way to run without templates is to set the date of the PDB to January 1, 1900 so that no templates end up being included in the calculations.

3) While I see the rationale for analyzing only the experimental structures through most of the paper, if AF2 (run without templates) produces good models of the experimental structures, it would be possible to merge the experimental and AF2 structures into one analysis. Perhaps the experimental structures

should be analyzed first. Then show that AF2 produces good models of those sequence pairs that are already in the PDB (i.e. such that analysis of the AF2 structures that are in the PDB produces the same results as using the experimental structures), and then conclude with an analysis of the joint set of experimental and AF2 structures. Of course, if the AF2 structures are not good enough, then that point could be made and the joint analysis might take on less significance. AF2 provides average pLDDTs, pTM, and iPTM scores that may be used to determine which models are reliable and which are not. The comparison with experimental structures can be used to set cutoffs.

4) An important distinction for GPCR structures is whether they are in an active or inactive conformation. GPCRdb has this information, and this should be annotated for the experimental structures in this study. The AF2 GPCR/Galpha complexes could also be annotated as active or inactive. Heo and Feig recently showed that AF2 prefers inactive structures of GPCRs (doi: <https://doi.org/10.1101/2021.11.26.470086>) but this may only be when the structure is predicted as a monomer.

5) How were the top AF2 models chosen? AF2 outputs average pLDDTs, pTM, and iPTM (interaction pTM). Probably iPTM is the best to use here since it is likely to have the best model of the interaction of the GPCR with the Galpha protein, which is the focus of this study. It would be interesting to know if there was any systematic reason why some AF2 structures did not pass the three criteria used in Table S4.

6) A lot of interesting data were generated in this paper. Since the github repository is not yet online, it was impossible to consider in reviewing the paper whether the authors are providing all the necessary data and annotating it in sufficient depth for it to be useful to people. As mentioned above, the AF2 models should be made available. But perhaps also the relaxed experimental structures would be useful to people too. Pymol or Chimera sessions for the figures (or whatever program was used to visualize structures – this is not mentioned in the paper) could be provided.

7) The paper should probably include an image of a typical GPCR/Galpha complex with the different secondary structures and loops annotated with their labels used in all the figures of this paper.

8) Minor:

a) Line 154: What is “(Figure 3A and Fig. 3A)” supposed to be? Fig S3A?

b) Glycine is consistently misspelled

c) The InterfaceAnalyzer citation misses the people who worked on that code the most including my former student Jared Adolf-Bryfogle. You could cite the Rosetta documentation:
https://www.rosettacommons.org/docs/latest/application_documentation/analysis/interface-analyzer

Reviewer #4 (Remarks to the Author):

In their analysis, the authors explore the highly relevant topic of coupling specificity of receptor G proteins. Despite recent progress in elucidating dozens of different receptor G protein and arrestin complexes, coupling specificity is still not fully understood. In their current analysis, the authors extend the existing dataset of available Gi/o and Gs receptor complexes with models provided by AlphaFold2 to obtain a more comprehensive dataset of these complexes. As a result, they find different binding energies, estimated by Rosetta InterfaceAnalyzer for Gi/o compared to Gs complexes. Overall, the analysis confirms previous works.

The present work uses a current tool to get a more comprehensive view of nucleotide-free receptor G protein complexes. Although the contact fingerprints provide detailed statistics of the different binding modes of Gs and Gi/o coupled complexes, and furthermore albeit to a lesser extent to Gq and G12/13, comparatively little new insight is provided beyond the previous understanding of receptor G protein coupling specificity. One reason for this may be the fact that the modeled complexes follow the scheme of nucleotide-free complexes, which have already been extensively analyzed. Moreover, these states have been stabilized under nucleotide-free preparations and do not necessarily represent the physiologically relevant interaction space (Du et al. and Kobilka, CELL, 2019).

An in-depth comparison with state-of-the-art knowledge, represented for instance by comparison of class A and B coupled with G protein is also lacking,

- Hauser, A. S. et al. Common coupling map advances GPCR-G-protein selectivity.

Elife 11, (2022)

- Hilger, D. et al. Structural insights into differences in G-protein activation by family A and family B GPCRs. Science (80-.). 369, (2020).

- Flock, T. et al. Universal allosteric mechanism for G α activation by GPCRs. Nature .

Thus, it is already known that H5 helix is the major player at the interface between G protein and GPCRs (Flock, T. et al.). Likewise, it is already known that Gi and Gs have different binding modes (Hauser, A. S. et al.).

Some of the illustrations seem a bit very small, in any case difficult to read especially the contact maps.

Reviewer #1 (Remarks to the Author)

In their manuscript „GPCRome-wide structural analysis of G-protein-coupling selectivity” the authors report an extensive computational study that is intended to shed light on the still largely unresolved question of G-protein-coupling selectivity. In brief, the authors use all currently available 3D GPCR/G-protein complex structures and AlphaFold2 predictions as a data basis to analyze residue-residue contacts between receptors and G-proteins. These contacts are used in fingerprinting and clustering analyses to determine structural differences between coupling of different transducers to receptors. Moreover, the authors perform calculations of binding energies that suggest that Gs and Gi binding to receptors can be distinguished energetically, with Gs proteins forming more stable ternary complexes than Gi proteins.

This is an interesting albeit highly technical manuscript that, for non-specialists in the field and likely for many experimentalists, can be sometimes very hard to follow. There are three major issues with the manuscript that do not justify publication in Nature communications:

We would like to thank Reviewer #1 for his/her comments of overall appreciation of our computational studies and for the useful suggestions to improve it. We have tried to address most of the suggestions and requests, please find below our point-to-point replies that we hope you will find satisfactory.

1) To me as an experimentalist, the conceptual novelty of this manuscript over previously published work is not clear. There have been numerous papers published in the past on this topic (e.g. PMID: 28489817; PMID: 31855179; PMID: 33846507; PMID: 35302494) and I cannot easily judge the advancement of this work over previous papers. It would be important to the reader to integrate all previous knowledge into the concept of the present manuscript.

Thank you for the suggestions. We have now included these references and better discussed them in the Introduction as well as in the Discussion sections in light of the previous computational analysis aimed at elucidating the sequence as well as the structural determinants of coupling selectivity (lines 75-80 and 335-337). Briefly, Flock et al., Nature, 2017 (PMID: 28489817) performed a sequence-based co-evolutionary analysis to identify determinants of binding on both the G-protein as well as on the receptor side. In this purely computational study, the authors analyzed some of their predictions in the context of the only experimental complex structure available back then (i.e. PDB 3sn6). In another study, Seo et al., Sci. Rep., 2021 (PMID: 33846507) presents a co-evolutionary analysis of multiple sequence alignment of the members of 5-hydroxytryptamine and dopamine receptors. Also in this case, 3D structures of receptor-G-proteins complexes have been merely used to map the results of the sequence-based analysis. Zhou et al., eLife, 2019 (PMID: 31855179) combined in-silico and in-vitro analysis to focus on the intramolecular network of interacting residues mediating the conformational changes associated with class A receptor activation. In this case, the focus is more on the intramolecular activation mechanisms rather than on the transducer binding. Hauser et al., eLife,

2022 (PMID: 35302494) integrated and harmonized dataset of binary experimental GPCR-G-protein couplings, but it did not provide any insights on the sequence or structural determinants of binding specificity (please also see our reply to Reviewer #4). In contrast, our study focuses on the analysis of 3D GPCR-G-protein complex interfaces via standard structural bioinformatic techniques, such as interface contact analysis and structural superimposition. In this respect, it should be considered as an extension of previous analysis, such as our own ones (Inoue, Raimondi et al., Cell, 2019; Okamoto et al., Nature Structural & Molecular Biology, 2021, PMID: 34354246) or from other groups (i.e. Huang et al., Molecular Cell, 2022, PMID: 35714614). In addition to analyze hundreds of experimental complexes, we have also predicted via AF-multimer the structures of receptor-heterotrimeric G-protein, whose α -subunits have been experimentally shown to binary interact with receptors. This study therefore provides not only the largest structural dataset of GPCR-G-protein complexes analyzed thus far to our knowledge, but also the first residue-level, comparative analysis of the structural basis of coupling selectivity of the human GPCRome.

2) Another significant drawback of the study is the complete lack of any experimental validation of the authors' results. First, it is very intriguing that the authors can devise potentially unique residues/contacts that are exclusively responsible for coupling to either G_i or G_s proteins. To substantiate these remarkable conclusions, it is important to e.g. design receptor or G-protein mutants at those residues and to demonstrate the presence or absence of an effect on G-protein coupling. Second, by using a promiscuous receptor, mutation of those residues specific for G_i/G_s -coupling should allow for the design of novel DREADDs. This should be demonstrated. Third, the peculiar contact patterns for $G_{12/13}$ -coupling should also be explored experimentally to e.g. unravel novel receptor/ $G_{12/13}$ pairings. Without any experimental validation, I do not think that this manuscript provides a sufficient conceptual advance in the field.

We thank Reviewer #1 for suggesting several experiments to validate our computational findings. However, we believe that some of these are much beyond the realistic amount of work required for this study and could be left for future studies. Regarding the requests to design new DREADDs (particularly a new $G_{12/13}$ DREADD) based on this structural analysis, this is something that our groups are already pursuing in collaboration. However, given the complexity of the task, and the amount of computational and experimental work that it is already taking, we feel that this study is worthy a dedicated, follow-up paper.

We have nevertheless explored the Reviewer #1's first request at point 2, i.e. designing receptor or G-protein mutants at those contact positions specific for either G_s or $G_{i/o}$ groups and experimentally demonstrating the presence of an effect on G-protein coupling. In this respect, we have shortlisted the contact position pairs most enriched in G_s and $G_{i/o}$ complexes. We then mutated the contacting positions on CCKAR, which has been solved in complex with both G_s (PDB ID: 7ezk) and $G_{i/o}$ (PDB ID: 7ezh). To this end we employed the Rosetta Multistate Design protocol (PMID: 21754981) to predict the substitutions favouring one G-protein coupling (either G_s or $G_{i/o}$) while diminishing the other. In this respect, we carried out two sets of designs: in one hand we sought to retain G_s while removing $G_{i/o}$ couplings, and on the other we aimed at maintaining $G_{i/o}$ while reducing the G_s binding (Figure4A). We found that certain mutations were more recurrent in top designed sequences. Specifically, mutations of A303^{6,25}K, V311^{6,33}H, K375^{8,48}R and R376^{8,49}L were

predicted to reduce $G_{i/o}$ while retaining G_s binding, whereas mutations of S149^{ICL2.53A}, V151^{ICL2.55K}, K308^{6.30R} and K375^{8.48P} were predicted to reduce G_s while retaining $G_{i/o}$ binding. These were subsequently tested through the NanoBIT G-protein dissociation assay. Among the eight designed mutations, two (V311^{6.33H} and R376^{8.49V}) and one (K308^{6.30R}) were found to be G_s -biased and $G_{i/o}$ -biased, respectively, and the rest of the five had no effect on the G_s -vs- G_i balance. These data support the importance of the identified contacts in switching coupling preferences between these two G-proteins. We show these new results in a new figure (Figure 4, see also below) as well as we describe them in a new Results section entitled (“Switching G-protein selectivity through contact interface mutation”).

3) The computational exploration of receptor/G-protein-binding energies is certainly important and interesting. However, it would be important to have some kind of experimental validation, e.g. ITC measurements or similar. Given the larger contact interface of receptors with G_s , it feels intuitive that the binding affinities are higher than those of G_i -complexes. Please confirm experimentally. On another note, the

distribution of ΔG for G_s appears to be much wider than for G_i . This would mean that there are GPCR/ G_s complexes with a lower free binding energy than G_i complexes. Please comment.

We thank Reviewer #1 for the suggestion, but we think that ITC experiments would require an enormous amount of experimental work, which again we think might be out of the scope of the present study. To the best of our knowledge, we did not find literature references showing ITC analysis between GPCR and G-protein, but only between receptor and ligands or nanobodies (e.g. PMID: 23747362 or 27409812), suggesting technical difficulties of such experiment. Our updated analysis again shows a slightly greater separation than the original analysis between the distributions of the binding energies of G_s vs $G_{i/o}$ complexes (Figure 6A). The distribution of binding energies for G_s is indeed wider compared to $G_{i/o}$, likely due to receptor differences. In fact, the distributions of binding energies of G_s complexes grouped on the basis of GPCR class are significantly different, with Class A being characterized by lower energies than class B ones (Figure 6C). In general, we observe very few G_s complexes characterized by binding energies as low, or lower, than $G_{i/o}$. The fact that different G-protein complexes made by the same receptor always show more stable G_s complexes than the $G_{i/o}$ ones suggest that G-protein is chiefly causing the observed differences.

Reviewer #2 (Remarks to the Author)

This paper presents a useful analysis of all currently available structures of complexes of GPCRs with Galpha proteins. Such structural bioinformatics studies are important because they synthesize a great deal of data, and information scattered about 100s of scientific papers that would not be easily searched or collated by individual scientists. On the whole, I think Nature Communications should consider such papers for publication. This paper is well done and worth considering for publication.

The authors do a solid job of constructing their data set and presenting their statistical analysis of residue-residue interactions between GPCRs and Galpha proteins and how these differ between different Galpha families. They also examine binding energies of Galpha/GPCR complexes and how these differ by Galpha and GPCR classes. They also undertake the important task of using AlphaFold2 to predict the structures of all likely GPCR/Galpha complexes.

I have several suggestions for the authors to consider that might improve or extend their analysis, mostly to do with how the AlphaFold2 structures are utilized. The AF2 structures are an exciting part of this paper, potentially significantly extending our understanding of the whole GPCR/Galpha-ome, and they are given relatively short shrift in the paper. Unless the authors think they are not accurate enough, this section could be extended.

We would like to thank Prof. Dunbrack for appreciating our work and for providing extremely valuable suggestions, which we addressed in the revised manuscript version. Admittedly, the AlphaFold result section was a bit neglected previously. Encouraged by the positive comments, we have now repeated AF-multimer predictions, extensively validated them, expanded the dedicated section as well as given more emphasis to this analysis in the story. Please find below more details.

1) It is not clear if the AlphaFold2 predicted structures will be made publicly available. I think this should be a condition of publication. The coordinate files should be made available on some repository such as zenodo or the authors' github repository, and should be annotated properly in mmCIF format including sequence records and UniProt numbering of the coordinates (in "auth_seq_id"). If the authors download some models from the EBI AlphaFold2 site (<https://www.alphafold.ebi.ac.uk/>), they can see what records are there and mimic those.

We have now made public the github repository (https://github.com/raimondilab/GPCR_structure_analysis) and uploaded the complex structures there, annotating in the "auth_seq_id" the Uniprot residue numbering. We will also use the newly predicted complexes to replace older models in our Precogx webserver (<https://precogx.bioinfolab.sns.it/>).

2) It would make sense to present and analyze AF2 predictions of GPCR/Galpha structures that are already in the PDB and compare the AF2 predictions with the experimental structures. For this purpose, AF2 can be run without templates. AF2-multimer does not use heteromer templates anyway, but the results will be more applicable to structures without templates if the sequence query pairs with templates

are also run with AF2 without PDB templates. The easiest way to run without templates is to set the date of the PDB to January 1, 1900 so that no templates end up being included in the calculations.

Thank you for this suggestion. We ran AF2-multimer with and without templates for 125 GPCR-G-protein complexes with available experimental structures, against which we compared the distance of the predictions using standard distance metric for evaluation of PPI 3D complex predictions (e.g. DockQ). In particular, we found that predictions with templates gave slightly better results, assessed either via DockQ distance or fraction of native contacts (see Figure 7A,B). It must be emphasized however that even without templates AF-multimer's predictions achieve almost comparable performances, yielding an average DockQ score of 0.645 vs. 0.664 obtained for predictions with templates. We therefore re-ran AF-multimer prediction with templates, by updating to AF2-multimer version 2.3, as well as by inputting the sequences of the receptor as well as of the α , β , γ subunits of the heterotrimeric G-proteins.

3) While I see the rationale for analyzing only the experimental structures through most of the paper, if AF2 (run without templates) produces good models of the experimental structures, it would be possible to merge the experimental and AF2 structures into one analysis. Perhaps the experimental structures should be analyzed first. Then show that AF2 produces good models of those sequence pairs that are already in the PDB (i.e. such that analysis of the AF2 structures that are in the PDB produces the same results as using the experimental structures), and then conclude with an analysis of the joint set of experimental and AF2 structures. Of course, if the AF2 structures are not good enough, then that point could be made and the joint analysis might take on less significance. AF2 provides average pLDDTs, pTM, and iPTM scores that may be used to determine which models are reliable and which are not. The comparison with experimental structures can be used to set cutoffs.

Thank you for the suggestion. While we have kept the analysis of experimental structures in the first Results section, we have largely redesigned and rewritten the results section describing AF2-multimer predictions (see section “*AlphaFold2 predictions extend our understanding of the structural basis of coupling specificity.*” and Figure 7). We first show the performances of AF2-multimer comparing predicted with available experimental structures, employing DockQ and fraction of native contact as distance metrics (Figure 7A-B). Since we noticed that some predicted complexes showed unrealistic docking topology, we created a composite filter based on the following criteria: a) docking topology filter, to retain only those complexes showing characteristic H5 contacts that are always observed in experimental structures; b) pDockQ score greater than 0.23, according to recently established protocols (e.g. PMID: 35273146, 36690744); c) filtering out residues having pLDDT < 70; d) filtering of complexes whose dSASA (as calculated via InterfaceAnalyzer) is greater than 1500 Å², which we defined on the basis of the same analysis done on experimental structures. Such filtering scheme improves the correlation between prediction scores (pDockQ) and distance from experimental structures (DockQ) in the benchmark (Figure 7C,D). We applied this filter to multimer's predictions, yielding a set of 825 complexes for downstream processing (Table S5). We then present the results of the analysis carried out on this pool of predicted structures, where we also include the ones with available experimental structures. We present a comparative

contact analysis, using a novel heatmap representation, where each cell corresponds to an interface contact which contains the statistics derived from the complexes of each coupling group colored in a group-specific scheme manner (Figure 7E). In this master heatmap, we also mark with an asterisk those contacts observed in experimental structures. We conclude the analysis by showing the distribution of the binding energies calculated for the predicted complexes grouped based on the four G-protein families (Figure 7F). Intriguingly, we show that the complexes within each G-protein family are characterized by significantly different binding energy distributions. Indeed, in addition to confirm the difference in binding energies observed for experimental structures of G_s and $G_{i/o}$ complexes, we also show significantly different binding energies for the other groups, with the $G_{12/13}$ one being the class of least stable complexes together with $G_{i/o}$ ones (Figure 7F).

4) An important distinction for GPCR structures is whether they are in an active or inactive conformation. GPCRdb has this information, and this should be annotated for the experimental structures in this study. The AF2 GPCR/Galpha complexes could also be annotated as active or inactive. Heo and Feig recently showed that AF2 prefers inactive structures of GPCRs (doi: <https://doi.org/10.1101/2021.11.26.470086>) but this may only be when the structure is predicted as a monomer.

In this study we considered complexes between receptors and heterotrimeric G-proteins, which is generally known to occur upon ligand-mediated activation of the receptor. We have retrieved from GPCRdb the classification of the “State” of each structure and integrated it in Table S1. A total of 352, out of 362, complexes (97%) are classified as Active, 2 as Intermediate and for 8 the State is not available.

5) How were the top AF2 models chosen? AF2 outputs average pLDDTs, pTM, and iPTM (interaction pTM). Probably iPTM is the best to use here since it is likely to have the best model of the interaction of the GPCR with the Galpha protein, which is the focus of this study. It would be interesting to know if there was any systematic reason why some AF2 structures did not pass the three criteria used in Table S4.

We selected the top model according to the AF-multimer default score, which is given by a combination of PTM and iPTM scores through the following equation: $0.2 \cdot \text{PTM} + 0.8 \cdot \text{iPTM}$, which indeed over-weights iPTM. We also checked whether selecting the top model by using either the default score, rather than the pDockQ, affects the results of the structural benchmark, and we found no significant difference, both with templates:

6) A lot of interesting data were generated in this paper. Since the github repository is not yet online, it was impossible to consider in reviewing the paper whether the authors are providing all the necessary data and annotating it in sufficient depth for it to be useful to people. As mentioned above, the AF2 models should be made available. But perhaps also the relaxed experimental structures would be useful to people too. Pymol or Chimera sessions for the figures (or whatever program was used to visualize structures – this is not mentioned in the paper) could be provided.

We have now made public the github repository (https://github.com/raimondilab/GPCR_structure_analysis) where we uploaded the predicted complex structures, as well as the relaxed structures. We also provided ChimeraX and Pymol session files that we have used to generate the figures. Finally, we will also update the AF predicted complexes that we were already providing through our Precogx webserver (<https://precogx.bioinfolab.sns.it/>)

7) The paper should probably include an image of a typical GPCR/Galpha complex with the different secondary structures and loops annotated with their labels used in all the figures of this paper.

Done. We added a cylindrical cartoon representation of a prototypical GPCR-heterotrimeric G-protein complex, i.e. β_2 adrenergic receptor-Gs (PDB:3sn6), labelled with secondary structural elements (Figure 2A).

8) Minor:

a) Line 154: What is “(Figure 3A and Fig. 3A)” supposed to be? Fig S3A?

Fixed, retained only the reference to Figure 3D

b) Glycine is consistently misspelled

Fixed

c) The InterfaceAnalyzer citation misses the people who worked on that code the most including my former student Jared Adolf-Bryfogle. You could cite the Rosetta documentation: https://www.rosettacommons.org/docs/latest/application_documentation/analysis/interface-analyzer

Added the URL in the appropriate Methods sections

Reviewer #4 (Remarks to the Author)

In their analysis, the authors explore the highly relevant topic of coupling specificity of receptor G-proteins. Despite recent progress in elucidating dozens of different receptor G-protein and arrestin complexes, coupling specificity is still not fully understood. In their current analysis, the authors extend the existing dataset of available Gi/o and Gs receptor complexes with models provided by AlphaFold2 to obtain a more comprehensive dataset of these complexes. As a result, they find different binding energies, estimated by Rosetta InterfaceAnalyzer for Gi/o compared to Gs complexes. Overall, the analysis confirms previous works.

The present work uses a current tool to get a more comprehensive view of nucleotide-free receptor G-protein complexes. Although the contact fingerprints provide detailed statistics of the different binding modes of Gs and Gi/o coupled complexes, and furthermore albeit to a lesser extent to Gq and G12/13, comparatively little new insight is provided beyond the previous understanding of receptor G-protein coupling specificity. One reason for this may be the fact that the modeled complexes follow the scheme of nucleotide-free complexes, which have already been extensively analyzed. Moreover, these states have been stabilized under nucleotide-free preparations and do not necessarily represent the physiologically relevant interaction space (Du et al. and Kobilka, CELL, 2019).

We thank reviewer #4 for the valuable comments. The point regarding the modeling of nucleotide-free complexes is indeed worth exploring, and we have now added a sentence addressing it in the discussion section, as well as included the recommended references (lines 392-394). We agree that in our analysis we are likely missing several of the interactions that might mediate the early steps of complex recognition and formation. Another critical aspect, which we recognize is largely missed by this analysis, is the contribution to coupling specificity by residues distal from the G-protein binding interface and which might allosterically control the binding process, as we discussed in our earlier work (e.g. Inoue, Raimondi, et al., Cell, 2019). However, the goal of this study is to perform a systematic structural bioinformatic characterization, using standard tools and metrics, of available experimental structures, which we augmented through AF2-multimer predictions. We showed that unsupervised clustering on contact networks and complex RMSD can overall recapitulate G_s vs G_{i/o} binding preferences on experimental structures. These structural hallmarks are in turn linked to significantly different binding energies between G_s and G_{i/o} complex groups. This trend is not only observed by AF2-multimer predictions, but extended to the other groups, i.e. G_{q/11} and G_{12/13}, which are depleted in experimental structures, and show significantly different binding energies distribution for each group, with the latter characterized by the least stable complexes. To our knowledge, such a systematic, quantitative characterization is not available in the literature and could be of great interest and value to the community.

An in-depth comparison with state-of-the-art knowledge, represented for instance by comparison of class A and B coupled with G-protein is also lacking,

- Hauser, A. S. et al. Common coupling map advances GPCR-G-protein selectivity. Elife 11, (2022)

- Hilger, D. et al. Structural insights into differences in G-protein activation by family A and family B GPCRs. Science (80-.). 369, (2020).

- Flock, T. et al. *Universal allosteric mechanism for G α activation by GPCRs. Nature .*

Thus, it is already known that H5 helix is the major player at the interface between G-protein and GPCRs (Flock, T. et al.). Likewise, it is already known that Gi and Gs have different binding modes (Hauser, A. S. et al.).

Thank you for the informative comments. We have better integrated in the discussion of the current manuscript the suggested literature (see lines 335-337). However, we would like to emphasize that we had already commented the differences between class A and B receptor complexes already in the Abstract (lines 48-49), as well as in the Results, both at the level of the contact fingerprints (Figure 3F, lines 173-176), as well as at the level of the predicted binding energies for experimental structures (Figure 6C, lines 269-271), which we further discussed (lines 371-375) by citing the relevant literature (e.g. Hilger et al., Science, 2020). Our structural analysis is intended to generalize the analysis carried out by Hilger and co-workers on individual receptor systems (e.g. *ADRB2* and *GCGR*). While the common coupling map by Hauser and co-workers provide important insights regarding the different extent of selectivity of the different coupling groups, it is not a structural analysis and therefore misses the sequence and structural determinants of selective binding. Our analysis can be considered as an extension of the first structural determination of Gi/o complexes, which also included comparative analysis with Gs structures, (Draper-Joyce et al., Nature 2018, Garcia-Nafria et al., Nature, 2018, Kang et al., Nature, 2018, Koehl et al., Nature, 2018), as well as more recent, but still fragmentary analyses (Okamoto et al., Nature Struct. Mol. Biol., 2021 and Huang et al., Mol Cell., 2022), and it has the overarching goal to structurally illuminate GPCR-G-protein interaction experimental datasets such as the common coupling map.

Some of the illustrations seem a bit very small, in any case difficult to read especially the contact maps.

We strived to improve the quality of the figure 3 as well as of all the others wherever possible.

REVIEWERS' COMMENTS

Reviewer #1 (Remarks to the Author):

The authors have undertaken considerable effort to address the comments that I had raised to the previous version of this manuscript. However, I remain skeptical about the conceptual advance of this work mainly for two reasons:

1) Despite the detailed technical outline about how the presented work differs from previous computational work, it remains unclear how the actual knowledge gained from this study advances the field. I understand that a receptor/G protein-coupling analysis has never been performed on this scale, however, and somewhat unfortunately, the analysis provided here does not yield significant new conceptual insight into this fundamental question over previous work. I share the criticism of Reviewer 4 in this point.

2) The way they were designed and performed, the G-protein dissociation experiments do not allow to assess G-protein bias and, thus, do not support the conclusions of the authors. To quantify biased signaling of any sort, it is essential to provide concentration-response curves of all mutants in both assays (Gi and Gs). Simply comparing Emax values (even at similar receptor expression levels) is insufficient to assess potential signaling bias as it does not take into account potential changes in ligand affinity and efficacy at the various receptor mutants. In addition to the technical flaws and based on the data provided in the new Figure 4E, I would disagree with the authors that the mutations V3116.33H and R3768.49V lead to a Gs-biased receptor. The data in Figure 4E rather suggest a global loss in G-protein coupling that affects both Gi and Gs activation. The data with the K3086.30R mutation look more promising, however, due to the complete lack of statistics, one cannot judge whether there are significant differences in the various data sets.

Reviewer #2 (Remarks to the Author):

The authors have responded well to all of my comments and I recommend publication of the paper.

Reviewer #4 (Remarks to the Author):

This work completes the repertoire of nucleotide-free structures by applying alpha-fold modeling, emphasizing mainly what is already known. It is quite consistent to model previously experimentally

resolved structures based on the existing coordinates of receptor G-protein complexes deposited in the PDB. The authors do this task in an exemplary manner, and this also applies to the extensive analyses of the complexes and models. The only thing that still leaves me in doubt is the very limited gain in knowledge regarding coupling specificity.

The work is based on the assumption, or at least the analysis is so motivated, that the nucleotide-free structures provide insights into this very coupling specificity. This assumption is now considered to be more or less disproved. The focus on the nucleotide-free structures for many years is merely due to the fact that these structures, which actually represent only a very short-lived transition state in the cell, were the only ones that were experimentally accessible at all - by removing all nucleotides after hours of incubation with apyrase - and produced extreme energy minimum states that are so unlikely to occur in the cell. However, coupling specificity will certainly be tested before GDP release.

Recently, Lambert and colleagues have shown (<https://pubmed.ncbi.nlm.nih.gov/36646958/>) that selectivity degrades when the release of nucleotides is not required for the formation of a GPCR-G protein complex, to the extent that most GPCRs interact with most nucleotide-coupled G proteins. In terms of potential insights into coupling specificity, therefore, it makes no significant difference whether more are added to the many dozens of experimentally determined nucleotide-free complexes, the significance of which is largely derived from detailed mapping of ligand-binding pockets. In this respect at least, the title seems misleading.

In summary, the present work, which is certainly technically "state of the art," basically takes full advantage of the limited view that nucleotide-free structures provide on coupling specificity by including the complete GPCRome now. That the gain in knowledge remains low was to be expected with this approach.

Reviewer #1 (Remarks to the Author):

The authors have undertaken considerable effort to address the comments that I had raised to the previous version of this manuscript. However, I remain skeptical about the conceptual advance of this work mainly for two reasons:

1) Despite the detailed technical outline about how the presented work differs from previous computational work, it remains unclear how the actual knowledge gained from this study advances the field. I understand that a receptor/G protein-coupling analysis has never been performed on this scale, however, and somewhat unfortunately, the analysis provided here does not yield significant new conceptual insight into this fundamental question over previous work. I share the criticism of Reviewer 4 in this point.

We thank reviewer #1 for her/his constructive criticism.

We recognize that coupling selectivity is a complex, dynamic process that is also determined by the early recognition events leading to the formation of a transient complex before nucleotide release. The process might even be regulated allosterically by distant residues, which are aspects that we are not considering in this study. Admittedly, our unsupervised clustering based on structural descriptors of the nucleotide-free state is not able to fully recapitulate coupling preferences, although we can find strong G protein coupling enrichments in each cluster (see below). We have therefore toned down the references to selectivity both in the title, now “GPCRome-wide analysis of G-protein-coupling diversity using a computational biology approach” as well as throughout the text (see for instance lines 394-402).

While some of the results confirm previous studies (e.g. role played by TM5 and TM6 in switching the coupling preference for G_s and $G_{i/o}$), other insights are not, such as predicted interface binding energies, at least to our knowledge. The clustering of the complexes based on interface contacts or RMSDs from structural superpositions shows, in an unsupervised fashion, that structural characteristics of the interface are good descriptors of the coupling specificity, especially for G_s and $G_{i/o}$ coupling groups. This allows us to identify clusters of complexes that are enriched in specific experimental couplings (Figures 3B and 5B). Such an unsupervised, data-driven approach to our knowledge has never been adopted to analyze GPCR-G protein 3D complexes. Furthermore, this study suggests that the structural descriptors of the interface of nucleotide-free complexes are informative, at least partially, about the coupling specificity.

Interface binding energy predictions add a new perspective to this picture, suggesting that structural characteristics are linked to different energies of binding, and different G protein coupling groups are characterized by different binding energies. This computational analysis is even more valuable, considering the difficulty in getting such information experimentally.

Our computational study has generated multiple hypotheses regarding the determinants of coupling selectivity, observed in the nucleotide free complex, that can be experimentally verified. Following the reviewer #1's previous suggestions, we had included validation of some of the computational findings of our study, which hopefully we, and the community, will continue to explore in follow-up studies.

2) The way they were designed and performed, the G-protein dissociation experiments do not allow to assess G-protein bias and, thus, do not support the conclusions of the authors. To quantify biased signaling of any sort, it is essential to provide concentration-response curves of all mutants in both assays (G_i and

G_s). Simply comparing E_{max} values (even at similar receptor expression levels) is insufficient to assess potential signaling bias as it does not take into account potential changes in ligand affinity and efficacy at the various receptor mutants. In addition to the technical flaws and based on the data provided in the new Figure 4E, I would disagree with the authors that the mutations V3116.33H and R3768.49V lead to a G_s-biased receptor. The data in Figure 4E rather suggest a global loss in G-protein coupling that affects both G_i and G_s activation. The data with the K3086.30R mutation look more promising, however, due to the complete lack of statistics, one cannot judge whether there are significant differences in the various data sets.

In the NanoBiT-G-protein assay, we measured responses to titrated ligand concentrations. In the previous revision, we only showed the E_{max} bar graph. In the second revision, we included the ΔpEC₅₀ bar graph, which was calculated from the same experiment datasets as the E_{max} graph. In addition, we performed statistical analysis (multiple paired t-test). In both of the parameters, we consistently observed that two (V311H and R376L) of the four G_s-positive design mutants show G_s-preferring responses while one (K308A) of the G_i-positive design mutants shows the opposite preference. We have modified Figure 4 by including all the necessary statistics. As for the global loss vs. specific loss of the G-protein activation for V311H and R376L, we agree that the most apparent effect in V311H is reduction of both G_s and G_i activity. However, as shown in the E_{max} and ΔpEC₅₀ bar graphs, G_i activity was affected more than G_s. For R376L, when comparing with WT (1:4), the expression-matched condition, G_s activity was retained (E_{max} bar graph) while G_i activity was reduced. Therefore, we consider that the three mutants became either G_s-over-G_i biased (V311H, R376L) or G_i-over-G_s biased (K308A), as designed. Please also see the updated Figure 4E-G.

Reviewer #2 (Remarks to the Author):

The authors have responded well to all of my comments and I recommend publication of the paper.

We thank one more time Reviewer #2, Prof. Dunbrack, for the very constructive comments which helped to substantially improve the manuscript.

Reviewer #4 (Remarks to the Author):

This work completes the repertoire of nucleotide-free structures by applying alpha-fold modeling, emphasizing mainly what is already known. It is quite consistent to model previously experimentally resolved structures based on the existing coordinates of receptor G-protein complexes deposited in the PDB. The authors do this task in an exemplary manner, and this also applies to the extensive analyses of the complexes and models. The only thing that still leaves me in doubt is the very limited gain in knowledge regarding coupling specificity.

The work is based on the assumption, or at least the analysis is so motivated, that the nucleotide-free structures provide insights into this very coupling specificity. This assumption is now considered to be more or less disproved. The focus on the nucleotide-free structures for many years is merely due to the fact that these structures, which actually represent only a very short-lived transition state in the cell, were the only ones that were experimentally accessible at all - by removing all nucleotides after hours of incubation with

apyrase - and produced extreme energy minimum states that are so unlikely to occur in the cell. However, coupling specificity will certainly be tested before GDP release.

Recently, Lambert and colleagues have shown (<https://pubmed.ncbi.nlm.nih.gov/36646958/>) that selectivity degrades when the release of nucleotides is not required for the formation of a GPCR-G protein complex, to the extent that most GPCRs interact with most nucleotide-coupled G proteins. In terms of potential insights into coupling specificity, therefore, it makes no significant difference whether more are added to the many dozens of experimentally determined nucleotide-free complexes, the significance of which is largely derived from detailed mapping of ligand-binding pockets. In this respect at least, the title seems misleading.

In summary, the present work, which is certainly technically "state of the art," basically takes full advantage of the limited view that nucleotide-free structures provide on coupling specificity by including the complete GPCRome now. That the gain in knowledge remains low was to be expected with this approach.

We thank reviewer #4 for her/his constructive criticism and for pointing to the recent work by the Lambert's group. We are now referring to it and commenting it in the discussion (lines 395-402). In this study mutant G proteins were used to mimic receptor-bound G proteins and to infer what likely occurs during the process of GPCR-G protein coupling. We note that in their analysis they did not directly consider the intermediate-state complexes that are suggested to play a role in coupling selectivity, but rather indirectly assessed the importance of such complexes by bypassing them. These results are consistent with a model where coupling selectivity is determined at several steps during receptor-catalyzed nucleotide exchange, at multiple sites over an evolving receptor-G protein interface via receptor-specific mechanisms.

We certainly agree with such proposed mechanism, and recognize that coupling selectivity is a complex, dynamic process that is also determined by the early recognition events leading to the formation of a transient complex before nucleotide release. We have toned down the references to selectivity both in the title, now "GPCRome-wide analysis of G-protein-coupling diversity using a computational biology approach" as well as throughout the text (see lines 394-402), which we consider better highlighting the major findings of this study (see lines 39-40).

Although our analysis is focused on the end state of the multi-step, selective binding process, i.e. the nucleotide-free state, we show that several structural descriptors of these states, such as interface contact, RMSD and predicted binding energies, are informative of coupling preferences via statistical, unsupervised learning techniques. Indeed, clustering of the complexes based on interface contacts or RMSDs from structural superpositions shows in an unsupervised fashion that structural characteristics of the interface indeed are good descriptors of the coupling specificity, at least for G_s and $G_{i/o}$ coupling groups, indeed being able to identify clusters of complexes that are enriched in specific experimental couplings (Figures 3B and 5B). Likewise, binding energies of the interface are potent discriminator of the different coupling classes. The availability of a highly accurate, thoroughly assessed structural prediction tool such AlphaFold-Multimer prompted us to predict the structures of binary complexes of recent, large scale experimental binding datasets. This information is particularly valuable for classes such as $G_{q/11}$ or $G_{12/13}$, that are depleted among experimentally solved structures. On such an augmented pool of predicted structures, we applied the same statistical, unsupervised learning techniques to the structural descriptors employed for the experimental structures, confirming several features observed among experimental structures as well as revealing new patterns, such as the lower binding affinities of $G_{12/13}$ complexes.

Such a data-driven approach, where we apply for the first time statistical learning techniques to the largest pool of experimental and highly accurate predicted structures, is quite powerful in revealing, in an unbiased fashion, that indeed the structural complementarity of nucleotide-free interfaces is informative about coupling preferences. Although certainly not conclusive to solve the coupling selectivity issue, the structures and analysis of the current study could nevertheless be considered as a reference resource, since they provide detailed structural information of the nucleotide-free end state, which could help guiding future studies aiming at fully understanding the determinants of coupling specificities via characterization of earlier, intermediate states, for example by systematically comparing the conformations of receptors and G proteins in isolation and in complex.